# Rate-distortion theory of neural coding and its implications for working memory

Anthony MV Jakob[1,2]*, Samuel J Gershman[3,4]

[1]Section of Life Sciences Engineering, École Polytechnique Fédérale de Lausanne, Lausanne, Switzerland; [2]Department of Neurobiology, Harvard Medical School, Boston, United States; [3]Department of Psychology and Center for Brain Science, Harvard University, Cambridge, United States; [4]Center for Brains, Minds, and Machines, MIT, Cambridge, United Kingdom

**Abstract** Rate-distortion theory provides a powerful framework for understanding the nature of human memory by formalizing the relationship between information rate (the average number of bits per stimulus transmitted across the memory channel) and distortion (the cost of memory errors). Here, we show how this abstract computational-level framework can be realized by a model of neural population coding. The model reproduces key regularities of visual working memory, including some that were not previously explained by population coding models. We verify a novel prediction of the model by reanalyzing recordings of monkey prefrontal neurons during an oculo-motor delayed response task.

## Editor's evaluation

This important study describes a model neural circuit that learns to optimally represent its inputs subject to an information capacity limit. This novel hypothesis provides a bridge between the theoretical frameworks of rate-distortion theory and neural population coding. Convincing evidence is presented that this model can account for a range of empirical phenomena in the visual working memory literature.

*For correspondence:
anthony.jakob@outlook.com

**Competing interest:** The authors declare that no competing interests exist.

## Introduction

All memory systems are capacity limited in the sense that a finite amount of information about the past can be stored and retrieved without error. Most digital storage systems are designed to work without error. Memory in the brain, by contrast, is error-prone. In the domain of working memory, these errors follow well-behaved functions of set size, variability, attention, among other factors. An important insight into the nature of such regularities was the recognition that they may emerge from maximization of memory performance subject to a capacity limit or encoding cost (*Sims et al., 2012*; *Sims, 2015*; *van den Berg and Ma, 2018*; *Bates et al., 2019*; *Bates and Jacobs, 2020*; *Brady et al., 2009*; *Nassar et al., 2018*).

Rate-distortion theory (*Shannon, 1959*) provides a general formalization of the memory optimization problem (reviewed in *Sims, 2016*). The costs of memory errors are specified by a *distortion function*; the capacity of memory is specified by an upper bound on the mutual information between the inputs (memoranda) and outputs (reconstructions) of the memory system. Systems with higher capacity can achieve lower expected distortion, tracing out an optimal trade-off curve in the rate-distortion plane. The hypothesis that human memory operates near the optimal trade-off curve allows one to deduce several known regularities of working memory errors, some of which we describe below. Past work has studied rate-distortion trade-offs in human memory (*Sims et al., 2012*; *Sims,*

*2015*; *Nagy et al., 2020*), as well as in other domains such as category learning (*Bates et al., 2019*), perceptual identification (*Sims, 2018*), visual search (*Bates and Jacobs, 2020*), linguistic communication (*Zaslavsky et al., 2018*), and decision making (*Gershman, 2020*; *Lai and Gershman, 2021*).

Our goal is to show how the abstract rate-distortion framework can be realized in a neural circuit using population coding. As exemplified by the work of Bays and his colleagues, population coding offers a systematic account of working memory performance (*Bays, 2014*; *Bays, 2015*; *Bays, 2016*; *Schneegans and Bays, 2018*; *Schneegans et al., 2020*; *Taylor and Bays, 2018*; *Tomić and Bays, 2018*), according to which errors arise from the readout of a noisy spiking population that encodes memoranda. We show that a modified version of the population coding model implements the celebrated Blahut–Arimoto algorithm for rate-distortion optimization (*Blahut, 1972*; *Arimoto, 1972*). The modified version can explain a number of phenomena that were puzzling under previous population coding accounts, such as *serial dependence* (the influence of previous trials on performance; *Kiyonaga et al., 2017*).

The Blahut–Arimoto algorithm is parametrized by a coefficient that specifies the trade-off between rate and distortion. In our circuit implementation, this coefficient controls the precision of the population code. We derive a homeostatic learning rule that adapts the coefficient to maintain performance at the capacity limit. This learning rule explains the dependence of memory performance on the intertrial and retention intervals (RIs) (*Shipstead and Engle, 2013*; *Souza and Oberauer, 2015*; *Bliss et al., 2017*). It also makes the prediction that performance should adapt across trials to maintain a set point close to the channel capacity. We confirm these performance adjustments empirically. Finally, we show that variations in performance track changes in neural gain, consistent with our theory.

## Results

### The channel design problem

We begin with an abstract characterization of the channel design problem, before specializing it to the case of neural population coding. A communication channel (*Figure 1A*) is a probabilistic mapping, $Q(\hat{\theta}|\theta)$, from input $\theta$ to a reconstruction $\hat{\theta}$. The input and output spaces are assumed to be discrete in our treatment (for continuous variables like color and orientation, we use discretization into a finite number of bins; see also *Sims, 2015*). We also assume that there is some capacity limit $C$ on the amount of information that this channel can communicate about $\theta$, as quantified by the mutual information $I(\theta; \hat{\theta})$ between $\theta$ and the stimulus estimate $\hat{\theta}$ decoded from the population activity. We will refer to $R \equiv I(\theta; \hat{\theta})$ as the channel's *information rate*. To derive the optimal channel design, we also need to specify what *distortion function* $d(\theta, \hat{\theta})$ the channel is optimizing—that is, how errors are quantified. Details on our choice of distortion function can be found below.

With these elements in hand, we can define the channel design problem as finding the channel $Q^*$ that minimizes expected distortion $D \equiv \mathbb{E}[d(\theta, \hat{\theta})]$ subject to the constraint that the information rate $R$ cannot exceed the capacity limit $C$:

$$Q^* = \arg\min_{Q:R \leq C} D. \tag{1}$$

For computational convenience, we can equivalently formulate this as an unconstrained optimization problem using a Lagrangian:

$$Q^* = \arg\min_{Q} R + \beta D, \tag{2}$$

where $\beta$ is a Lagrange multiplier equal to the negative slope of the rate-distortion function at the capacity limit:

$$\beta = -\frac{\partial R}{\partial D}. \tag{3}$$

Intuitively, the Lagrangian can be understood as expressing a cost function that captures the need to both minimize distortion (i.e., memory should be accurate) and minimize the information rate (i.e., memory resources should economized). The Lagrange parameter $\beta$ determines the trade-off

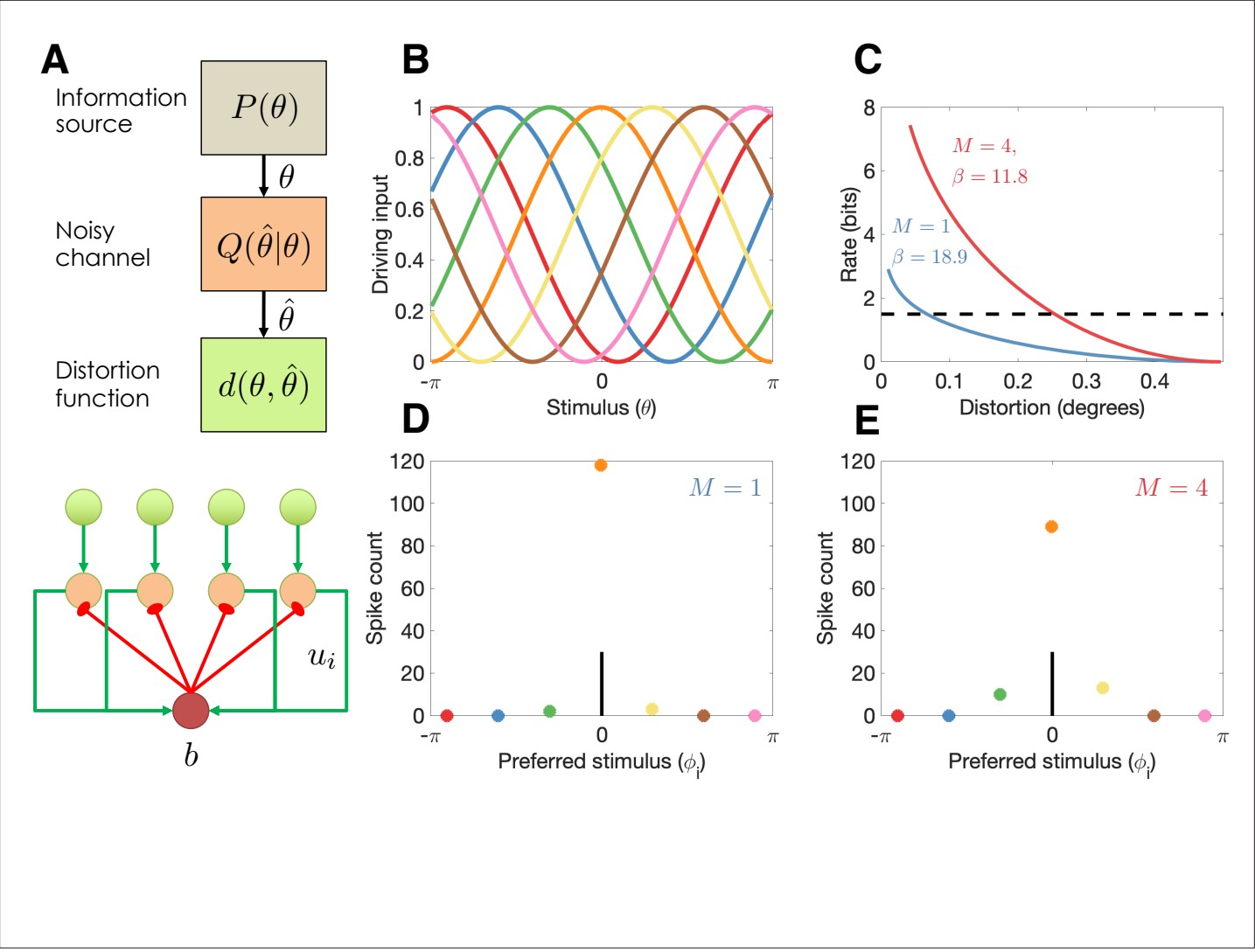

**Figure 1.** Model illustration. (**A**) Top: Abstract characterization of a communication channel. A stimulus $\theta$ is sampled from an information source $P(\theta)$ and passed through a noisy communication channel $Q(\hat{\theta}|\theta)$, which outputs a stimulus reconstruction $\hat{\theta}$. The reconstruction error is quantified by a distortion function, $d(\theta, \hat{\theta})$. Bottom: Circuit architecture implementing the communication channel. Input neurons encoding the negative distortion function provide the driving input to output neurons with excitatory input $u_i$ and global feedback inhibition $b$. Each circuit codes a single stimulus at a fixed retinotopic location. When multiple stimuli are presented, the circuits operate in parallel, interacting only through a common gain parameter, $\beta$. (**B**) Tuning curves of input neurons encoding the negative cosine distortion function over a circular stimulus space. (**C**) Rate-distortion curves for two different set sizes ($M = 1$ and $M = 4$). The optimal gain parameter $\beta$ is shown for each curve, corresponding to the point at which each curve intersects the channel capacity (horizontal dashed line). Expected distortion decreases with the information rate of the channel, but the channel capacity imposes a lower bound on expected distortion. (**D**) Example spike counts for output neurons in response to a stimulus ($\theta = 0$, vertical line). The output neurons are color coded by their corresponding input neuron (arranged horizontally by their preferred stimulus, $\phi_i$ for neuron $i$; full tuning curves are shown in panel B). When only a single stimulus is presented ($M = 1$), the gain is high and the output neurons report the true stimulus with high precision. (**E**) When multiple stimuli are presented ($M = 4$), the gain is lower and the output has reduced precision (i.e., sometimes the wrong output neuron fires).

between these two terms. Note that because the optimal trade-off function is always monotonically non-increasing and convex, the value of $\beta$ is always positive and non-increasing in $D$.

By integrating the ordinary differential equation defined in *Equations 2 and 3* and using the Lagrangian formulation, one can show that the optimal channel for a discrete stimulus takes the following form:

$$Q^*(\hat{\theta}|\theta) \propto \exp[-\beta d(\theta, \hat{\theta}) + \log \bar{Q}(\hat{\theta})], \tag{4}$$

where the marginal probability $\bar{Q}(\hat{\theta})$ is defined by:

$$\bar{Q}(\hat{\theta}) = \sum_{\theta} P(\theta)Q^*(\hat{\theta}|\theta).$$  (5)

These two equations are coupled. One can obtain the optimal channel by initializing them to uniform distributions and iterating them until convergence. This is known as the Blahut–Arimoto algorithm (*Blahut, 1972*; *Arimoto, 1972*).

For a channel with a fixed capacity $C$ but variable $D$ across contexts, the Lagrange multiplier $\beta$ will need to be adjusted for each context so that $R = C$. We can accomplish this by computing $R$ for a range of $\beta$ values and choosing the value that gets closest to the constraint $C$ (later we will propose a more biologically plausible algorithm). Intuitively, $\beta$ characterizes the sensitivity of the channel to the stimulus. When stimulus sensitivity is lower, the information rate is lower and hence the expected distortion is higher.

In general, we will be interested in communicating a collection of $M$ stimuli, $\theta = \{\theta_1, \ldots, \theta_M\}$, with associated probing probabilities $\pi = \{\pi_1, \ldots, \pi_M\}$, where $\pi_m$ is the probability that stimulus $m$ will be probed (*van den Berg and Ma, 2018*). The resulting distortion function is obtained by marginalizing over the probe stimulus:

$$d(\theta, \hat{\theta}) = \sum_{m} \pi_m d(\theta_m, \hat{\theta}_m).$$  (6)

## Optimal population coding

We now consider how to realize the optimal channel with a population of spiking neurons, each tuned to a particular stimulus (*Figure 1A*). The firing rate of neuron $i$ is determined by a simple Spike Response Model (*Gerstner and Kistler, 2002*) in which the membrane potential is the difference between the excitatory input, $u_i$, and the inhibitory input, $b$, which we model as common across neurons (to keep notation simple, we will suppress the time index for all variables). Spiking is generated by a Poisson process, with firing rate modeled as an exponential function of the membrane potential (*Jolivet et al., 2006*):

$$r_i = \exp[u_i - b].$$  (7)

We assume that inhibition is given by $b = \log \sum_i \exp[u_i]$, in which case the firing rate is driven by the excitatory input with divisive normalization (*Carandini and Heeger, 2011*):

$$r_i = \frac{\exp[u_i]}{\sum_j \exp[u_j]}.$$  (8)

The resulting population dynamics is a form of 'winner-take-all' circuit (*Nessler et al., 2013*). If each neuron has a preferred stimulus $\phi_i$, then the winner can be understood as the momentary channel output, $\hat{\theta} = \phi_i$ whenever neuron $i$ spikes (denoted $z_i = 1$). The probability that neuron $i$ is the winner within a given infinitesimal time window is:

$$q(\hat{\theta} = \phi_i|\theta) = r_i.$$  (9)

Importantly, *Equation 9* has the same functional form as *Equation 4*, and the two are equivalent if the excitatory input is given by:

$$u_i = -\beta d(\theta, \phi_i) + w_i,$$  (10)

where

$$w_i = \log \sum_{\theta} q(\hat{\theta} = \phi_i|\theta)P(\theta)$$  (11)

is the log marginal probability of neuron $i$ being selected as the winner. We can see from this expression that the first term in *Equation 10* corresponds to the neuron's stimulus-driven excitatory input and the second term corresponds to the neuron's excitability. The Lagrange multiplier $\beta$ plays the role of a gain modulation factor.

The excitability term can be learned through a form of intrinsic plasticity (**Nessler et al., 2013**), using the following spike-triggered update rule:

$$\Delta w_i = \eta \left( c \exp[-w_i] z_i - 1 \right), \tag{12}$$

where $\eta$ is a learning rate and $c$ a gain parameter. After a spike ($z_i = 1$), the excitability is increased proportionally to the inverse exponential of current excitability. In the absence of a spike, the excitability is decreased by a constant. This learning rule is broadly in agreement with experimental studies (**Daoudal and Debanne, 2003**; **Cudmore and Turrigiano, 2004**).

## Gain adaptation

We now address how to optimize the gain parameter $\beta$. We want the circuit to operate at the set point $R = C$, where the channel capacity $C$ is understood as some fixed property of the circuit, whereas the information rate $R$ can vary based on the parameters and input distribution, but cannot persistently exceed $C$. Assuming the total firing rate of the population is approximately constant across time, we can express the information rate as follows:

$$R = \mathbb{E}\left[ \log \frac{Q(\hat{\theta}|\theta)}{\bar{Q}(\hat{\theta})} \right] = \sum_{i=1}^{N} \sum_{\theta} P(\theta)\mathbb{E}[\log r_i|\theta] - \mathbb{E}[\log r_i], \tag{13}$$

where $N$ is the number of neurons. This expression reveals that channel capacity corresponds to a constraint on stimulus-driven deviations in firing rate from the marginal firing rate. When the stimulus-driven firing rate is persistently greater than the marginal firing rate, the population may incur an unsustainably large metabolic cost (**Levy and Baxter, 1996**; **Laughlin et al., 1998**). When the stimulus-driven firing rate is lower than the marginal firing rate, the population is underutilizing its information transmission resources. We can adapt the deviation through a form of homeostatic plasticity, by increasing $\beta$ when the deviation is below the channel capacity, and decreasing $\beta$ when the deviation is above the channel capacity. Concretely, a simple update rule implements this idea:

$$\Delta\beta = \alpha(C - R), \tag{14}$$

where $\alpha$ is a learning rate parameter. We assume that this update is applied continuously. A similar adaptive gain modulation has been observed in neural circuits (**Desai et al., 1999**; **Hengen et al., 2013**; **Hengen et al., 2016**). Mechanistically, this could be implemented by changes in background activity: when stimulus-driven excitation is high, the inhibition will also be high (the network is balanced), and the ensuing noise will effectively decrease the gain (**Chance et al., 2002**).

In this paper, we do not directly model how the information rate $R$ is estimated in a biologically plausible way. One possibility is that this is implemented with slowly changing extracellular calcium levels, which decrease when cells are stimulated and then slowly recover. This mirrors (inversely) the qualitative behavior of the information rate. More quantitatively, it has been posited that the relationship between firing rate and extracellular calcium level is logarithmic (**King et al., 2001**), consistent with the mathematical definition in *Equation 13*. Thus, in this model, capacity $C$ corresponds to a calcium set point, and the gain parameter adapts to maintain this set point. A related mechanism has been proposed to control intrinsic excitability via calcium-driven changes in ion channel conductance (**LeMasson et al., 1993**; **Abbott and LeMasson, 1993**).

## Multiple stimuli

In the case where there are multiple stimuli, the same logic applies, but now we calculate the information rate over all the subpopulations of neurons (each coding a different stimulus). Specifically, the excitatory input becomes:

$$u_{im} = -\beta\pi_m d(\theta_m, \phi_{im}) + w_{im}, \tag{15}$$

where $m$ indexes both stimuli and separate subpopulations of neurons tuned to each stimulus location (or other stimulus feature that individuates the stimuli). As a consequence, $\beta$ will tend to be smaller when more stimuli are encoded, because the same capacity constraint will be divided across more neurons.

## Memory maintenance

In delayed response tasks, the stimulus is presented transiently, and then probed after a delay. The channel thus needs to maintain stimulus information across the delay. Our model assumes that the excitatory input $u_i$ maintains a trace of the stimulus across the delay. The persistence of this trace is determined by the gain parameter $\beta$. Because persistently high levels of stimulus-evoked activity may, according to *Equation 13*, increase the information rate above the channel capacity, the learning rule in *Equation 14* will reduce $\beta$ and thereby functionally decay the memory trace.

The circuit model does not commit to a particular mechanism for maintaining the stimulus trace. A number of suitable mechanisms have been proposed (*Zylberberg and Strowbridge, 2017*). One prominent model posits that recurrent connections between stimulus-tuned neurons can implement an attractor network that maintains the stimulus trace as a bump of activity (*Wang, 2001*; *Amit and Brunel, 1997*). Other models propose cell-intrinsic mechanisms (*Egorov et al., 2002*; *Durstewitz and Seamans, 2006*) or short-term synaptic modifications (*Mongillo et al., 2008*; *Bliss and D'Esposito, 2017*). All of these model classes are potentially compatible with the theory that population codes are optimizing a rate-distortion trade-off, provided that the dynamics of the memory trace conform to the equations given above.

During time periods when no memory trace needs to be maintained, such as the intertrial interval (ITI) in delayed response tasks, we assume that the information rate is 0. Because the information rate is the *average* number of bits communicated across the channel, these 'silent' periods effectively increase the achievable information rate during 'active' periods (which we denote by $R_A$). Specifically, if $T_A$ is the active time (delay period length), and $T_S$ is the silent time (ITI length), then the channel's rate is given by:

$$R = \frac{T_A}{T_A + T_S} R_A. \tag{16}$$

Equivalently, we can ignore the intervals in our model and simply rescale the channel capacity by $(T_A + T_S)/T_A$. This will allow us to model the effects of delay and ITI on performance in working memory tasks.

## Implications for working memory

### Continuous report with circular stimuli

We apply the framework described above to the setting in which each stimulus is drawn from a circular space (e.g., color or orientation), $\theta_m \in (-\pi, \pi)$, which we discretize. Reconstruction errors are evaluated using a cosine distortion function:

$$d(\theta, \hat{\theta}) = -\omega \cos(\theta - \hat{\theta}), \tag{17}$$

where $\omega > 0$ is a scaling parameter. This implies that the input neurons have cosine tuning curves (*Figure 1B*), and the output neurons have Von Mises tuning curves, as assumed in previous population coding models of visual working memory for circular stimulus spaces (*Bays, 2014*; *Schneegans and Bays, 2018*; *Taylor and Bays, 2018*; *Tomić and Bays, 2018*). All of our subsequent simulations use the same tuning curves.

As an illustration of the model behavior in the continuous report task, we compare performance for set sizes 1 and 4. The optimal trade-off curves are shown in *Figure 1C*. For every point on the curve, the same information rate achieves a lower distortion for set size 1, due to the fact that all of the channel capacity can be devoted to a single stimulus (a hypothetical capacity limit is shown by the dashed horizontal line). In the circuit model, this higher performance is achieved by a narrow bump of population activity around the true stimulus (*Figure 1D*), compared to a broader bump when multiple stimuli are presented (*Figure 1E*).

In the following sections, we compare the full rate-distortion model (as described above) with two variants. The 'fixed gain' variant assumes that $\beta$ is held fixed to a constant (fit as a free parameter) rather than adjusted dynamically. The 'no plasticity' model holds the neural excitability to a fixed value (fit as a free parameter). These two variants remove features of the full rate-distortion model which critically distinguish it from the population coding model of working memory (*Bays, 2014*). As a

strong test of our model, we fit only to data from Experiment 1 in *Bays, 2014*, and then evaluated the model on the other datasets without fitting any free parameters.

## Set size

One of the most fundamental findings in the visual working memory literature is that memory precision decreases with set size (*Bays et al., 2009*; *Bays, 2014*; *Wilken and Ma, 2004*). Our model asserts that this is the case because the capacity constraint of the system is divided across more neurons as the number of stimuli to be remembered increases, thus reducing the recall accuracy for any one stimulus. *Figure 2A* shows the distribution of recall error for different set sizes as published in previous work (*Bays, 2014*). *Figure 2D* shows simulation results replicating these findings.

## Prioritization

Stimuli that are attentionally prioritized are recalled more accurately. For example, error variance is reduced by a cue that probabilistically predicts the location of the probed stimulus (*Bays, 2014*; *Yoo et al., 2018*). In our model, the cue is encoded by the probing probability $\pi_m$, which alters the expected distortion. This results in greater allocation of the capacity budget to cued stimuli than to uncued stimuli. *Figure 2B, C* shows empirical findings (variance and kurtosis), which are reproduced by our simulations shown in *Figure 2E, F*. Kurtosis is one way of quantifying deviation from normality: values greater than 0 indicate tails of the error distribution that are heavier than expected under a normal distribution. The 'excess' kurtosis observed in our model is comparable to that observed by Bays in his population coding model (*Bays, 2014*) when gain is sufficiently low. This is not surprising, given the similarity of the models.

## Timing

It is well established that memory performance typically degrades with the RI (*Pertzov et al., 2017*; *Panichello et al., 2019*; *Schneegans and Bays, 2018*; *Zhang and Luck, 2009*), although the causes of this degradation are controversial (*Oberauer et al., 2016*), and in some cases the effect is unreliable (*Shin et al., 2017*). According to our model, this occurs because long RIs tax the information rate of the neural circuit. In order to stay within the channel capacity, the circuit reduces the gain parameter $\beta$ for long RIs, thereby reducing the information rate and degrading memory performance.

Memory performance also depends on the ITI, but in the opposite direction: longer ITIs improve performance (*Souza and Oberauer, 2015*; *Shipstead and Engle, 2013*). The critical determinant of performance is in fact the ratio between the ITI and RI. *Souza and Oberauer, 2015* found that performance in a color working memory task was similar when both intervals were short or both intervals were long. They also reported that a *longer* RI could produce *better* memory performance when it is paired with a longer ITI. *Figure 3* shows a simulation of the same experimental paradigm, reproducing the key results. This timescale invariance, which is also seen in studies of associative learning (*Balsam and Gallistel, 2009*), arises as a direct consequence of *Equation 16*. Increasing the ITI reduces the information rate, since no stimuli are being communicated during that time period, and can therefore compensate for longer RIs.

## Serial dependence

Working memory recall is biased by recent stimuli, a phenomenon known as *serial dependence* (*Fischer and Whitney, 2014*; *Fritsche et al., 2017*; *Bliss et al., 2017*; *Papadimitriou et al., 2015*). Recall is generally attracted toward recent stimuli, though some studies have reported repulsive effects when the most recent and current stimulus differ by a large amount (*Barbosa et al., 2020*; *Bliss et al., 2017*). Our theory explains serial dependence as a consequence of the marginal firing rate of the output cells, which biases the excitatory input $u_i$ (see *Equation 10*). Because the marginal firing rate is updated incrementally, it will reflect recent stimulus history.

An important benchmark for theories of serial dependence is the finding that it increases with the RI and decreases with ITI (*Bliss et al., 2017*). These twin dependencies are reproduced by our model (*Figure 4*). Our explanation of serial dependence is closely related to our explanation of timing effects on recall error: the strength of serial dependence varies inversely with the information rate, which

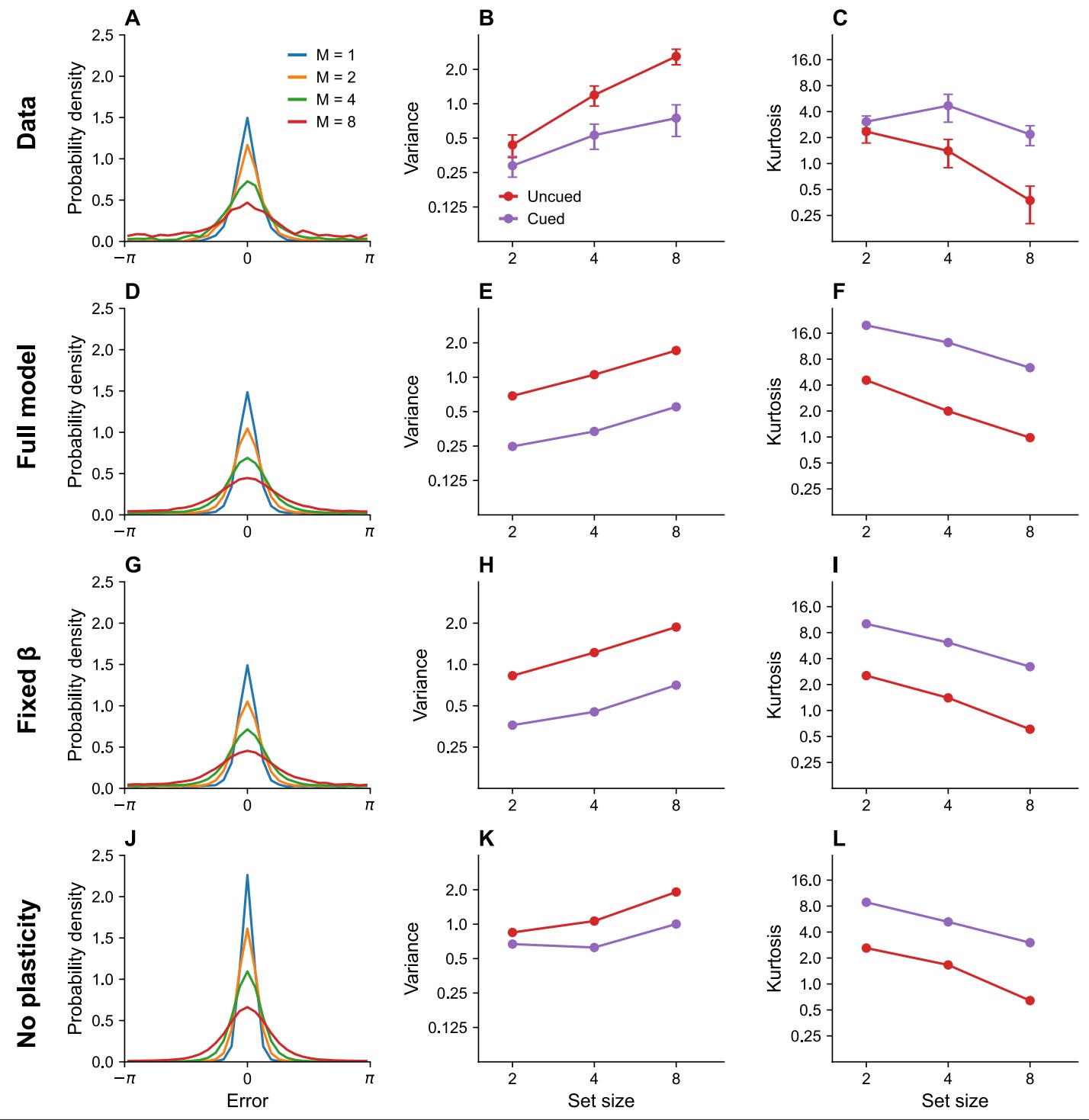

**Figure 2.** Set size effects and prioritization. (**A**) Error distributions for different set sizes, as reported in *Bays, 2014*. Error variability increases with set size. (**B**) Error variance as a function of set size for cued and uncued stimuli. Reports for cued stimuli have lower error variance. (**C**) Kurtosis as a function of set size for cued and uncued stimuli. Simulation results for the full model (**D–F**), model with fixed gain parameter $\beta$ (**G–I**), and model without plasticity term $w$ (**J–L**). Error bars represent standard error of the mean.

in turn increases with the ITI and decreases with the RI. Mechanistically, this effect is mediated by adjustments of the gain parameter $\beta$ in order to keep the information rate near the channel capacity. Serial dependence has also been shown to build up over the course of an experimental session (*Barbosa and Compte, 2020*). This is hard to explain in terms of theories based on purely short-term

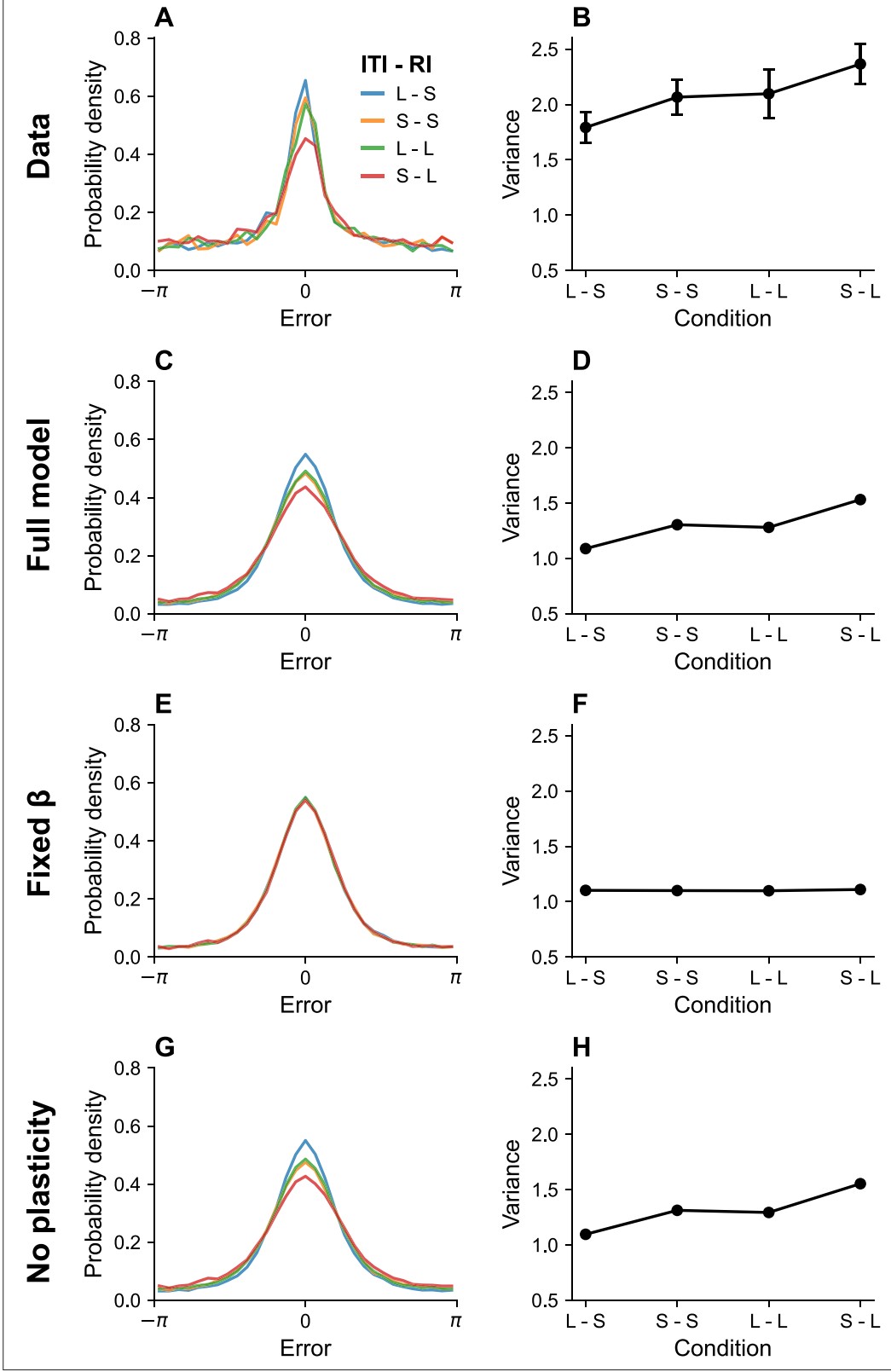

**Figure 3.** Timing effects. (**A**) Error distributions for different intertrial intervals (ITIs) and retention intervals (RIs), as reported in *Souza and Oberauer, 2015*. 'S' denotes a short interval, and 'L' denotes a long interval. (**B**) Error variance as a function of timing parameters. Longer ITIs are associated with lower error variance, whereas longer RIs are associated with larger error variance. Simulation results for the full model (**C, D**), model with fixed gain parameter $\beta$ (**E, F**), and model without plasticity term $w$ (**G, H**). Error bars represent standard error of the mean.

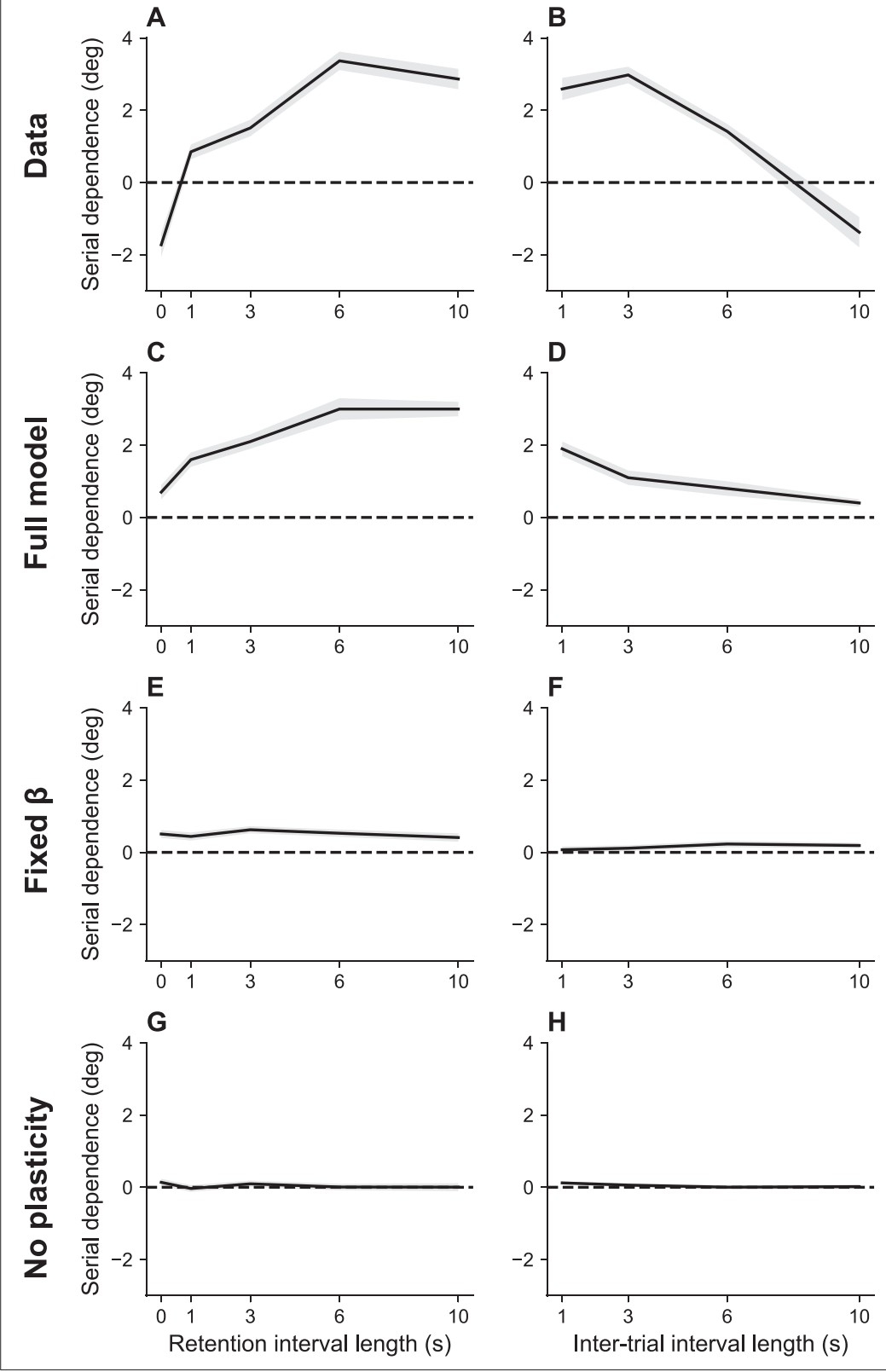

**Figure 4.** Serial dependence as a function of retention interval and intertrial interval. (**A**) Serial dependence increases with the retention interval until eventually reaching an asymptote, as reported in **Bliss et al., 2017**. Serial dependence is quantified as the peak-to-peak amplitude of a derivative of Gaussian (DoG) tuning function fitted to the data using least squares (see Methods). (**B**) Serial dependence decreases with intertrial interval. Simulation

*Figure 4 continued on next page*

*Figure 4 continued*

results for the full model (**C, D**), model with fixed gain parameter $\beta$ (**E, F**), and model without plasticity term $w$ (**G, H**). Shaded area corresponds to standard error of the mean.

effects, but it is consistent with our account in terms of the bias induced by the marginal firing rate. Because this bias reflects continuous incremental adjustments, it integrates over the entire stimulus history, thereby building up over the course of an experimental session (*Figure 5*).

If, as we hypothesize, serial dependence reflects a capacity limit, then we should expect it to increase with set size, since $\beta$ must decrease to stay within the capacity limit. To the best of our knowledge, this prediction has not been tested. We confirmed this prediction for color working memory using a large dataset reported in *Panichello et al., 2019*. *Figure 6* shows that the attractive bias for similar stimuli on consecutive trials is stronger when the set size is larger ($p < 0.05$, group permutation test).

## Systematic biases

Working memory exhibits systematic biases toward stimuli that are shown more frequently than others (*Panichello et al., 2019*). Moreover, these biases increase with the RI, and build up over the course of an experimental session. Our interpretation of serial dependence, which also builds up over the course of a session, suggests that these two phenomena may be linked (see also *Tong and Dubé, 2022*).

Our theory posits that, over the course of the experiment, the marginal firing rate asymptotically approaches the distribution of presented stimuli (assuming there are no inhomogeneities in the distortion function). Thus, the neurons corresponding to high-frequency stimuli become more excitable than others and bias recall toward their preferred stimuli. This bias is amplified by lower effective capacities brought about by longer RIs. *Figure 7* shows simulation results replicating these effects.

## Quantitative model comparison

To systematically compare the performance of the different models, we carried out random-effects Bayesian model comparison (*Rigoux et al., 2014*) for each dataset (see Methods). This method estimates a population distribution over models from which each subject's data are assumed to be sampled. The protected exceedance probabilities, shown in *Table 1*, quantify the posterior probability that each model is the most frequent in the population, taking into account the possibility of the null hypothesis where model probabilities are uniform.

Experiment 1 from *Bays, 2014* did not discriminate strongly between models. All the other datasets provided moderate to strong evidence in favor of the full rate-distortion model, with an average protected exceedance probability of 0.76.

## Variations in gain

*Equation 14* predicts that operating below the channel capacity will lead to an increase in the gain term $\beta$, which, in turn, leads to a higher information rate and better memory performance. Therefore, our model predicts that recall accuracy should improve after a period of poor memory performance, and degrade after a period of good memory performance. At the neural level, the model predicts that error will tend to be lower when gain ($\beta$) is higher.

We tested these predictions by reanalyzing the monkey neural and behavioral data reported in *Barbosa et al., 2020* ($N = 2$). The neural data were collected from the dorsolateral prefrontal cortex, a region classically associated with maintenance of information in working memory (*Levy and Goldman-Rakic, 2000*; *Funahashi, 2006*; *Wimmer et al., 2014*).

Behavioraly, squared error was significantly lower following higher-than-average error than following lower-than-average error (linear mixed model, $p < 0.001$; *Figure 8A*), consistent with the hypothesis that gain tends to increase after poor performance and decrease after good performance.

In order to estimate the neural gain, we first inferred the preferred stimulus of each neuron by fitting a bell-shaped tuning function to its spiking behavior (*Equation 23*, *Figure 8B*). We then performed Poisson regression to fit a $\beta$ for each neuron (*Equation 24*). Model comparison using the Bayesian information criterion (BIC) established that both the distortion function (which captures

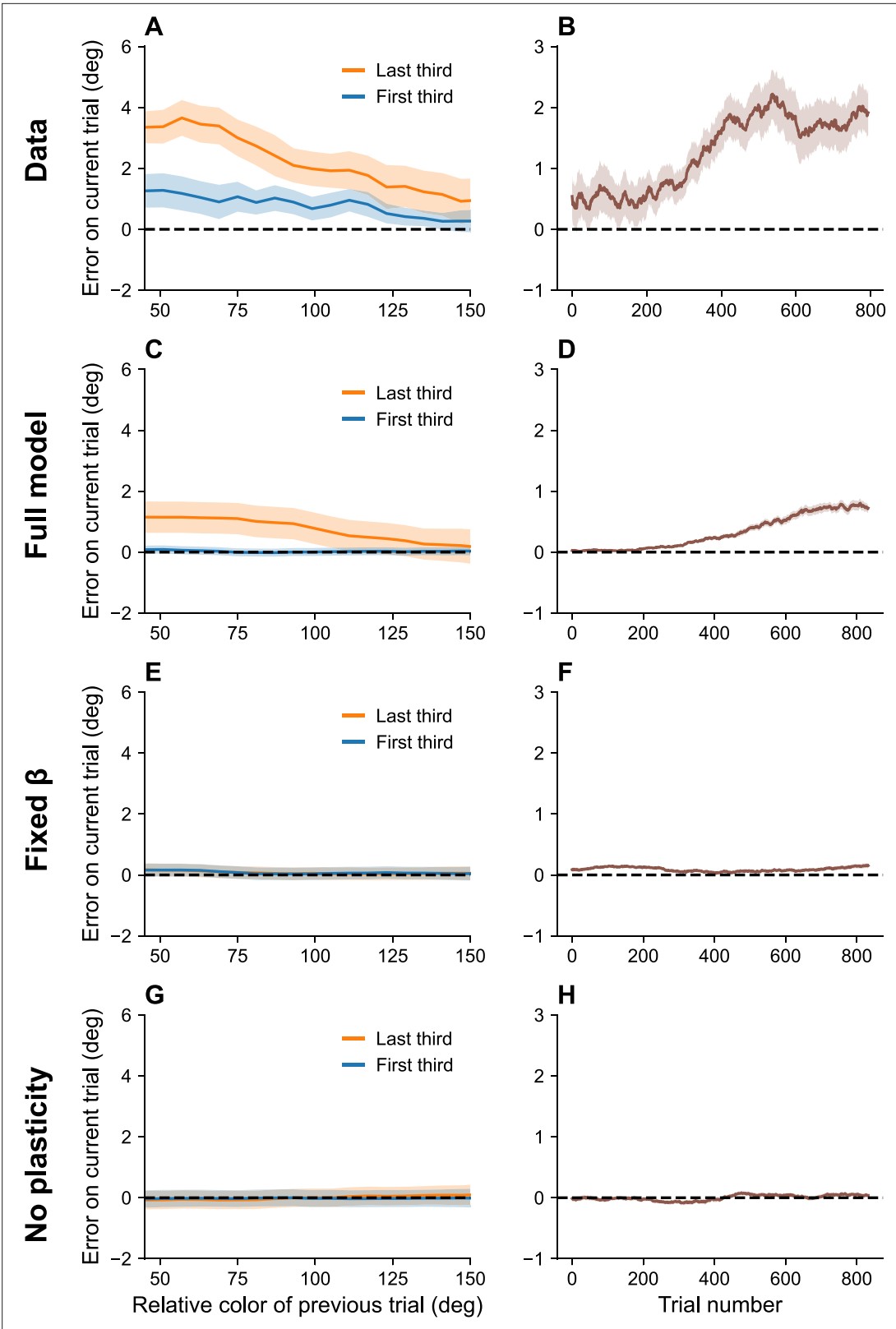

**Figure 5.** Serial dependence builds up during an experiment. (**A**) Serial dependence computed using first third (blue) and last third (orange) of the trials within a session, as reported in *Barbosa and Compte, 2020*. Data shown here were originally reported in *Foster et al., 2017*. To obtain a trial-by-trial measure of serial dependence, we calculated the folded error as described in *Barbosa and Compte, 2020* (see Methods). Positive values indicate attraction to the last stimulus, while negative values indicate repulsion. Serial dependence is stronger in the last third of the trials in the experiment

*Figure 5 continued on next page*

driving input) and spiking history were significant predictors of spiking behavior (full model: 54,545; no history: 59,163; neither distortion nor history: 67,903). We then examined the relationship between neural gain and memory precision across sessions, finding that session-specific mean squared error

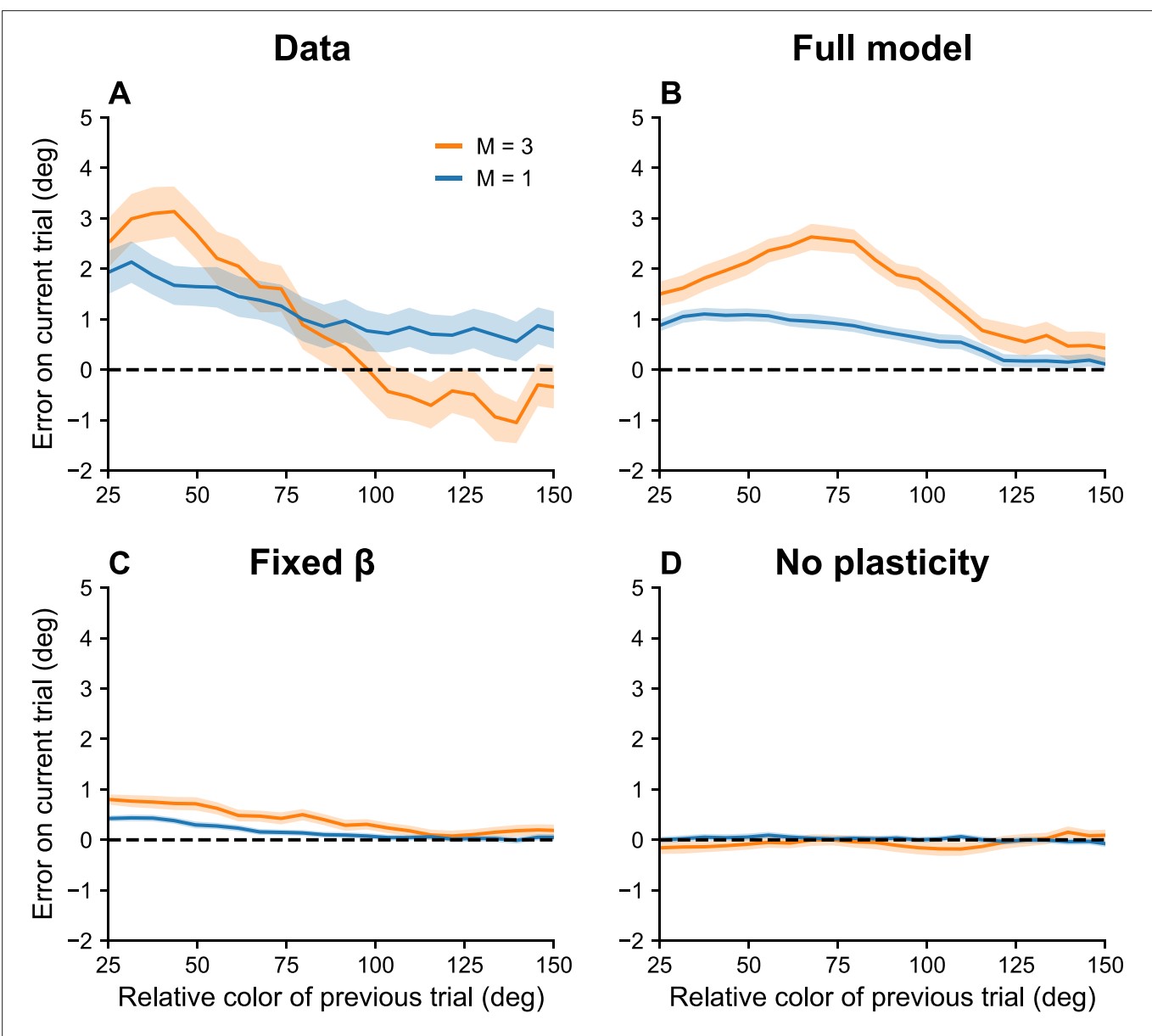

**Figure 6.** Serial dependence increases with set size. (**A**) Serial dependence (quantified using folded error) for set sizes $M = 1$ (blue) and $M = 3$ (orange), using data originally reported in *Panichello et al., 2019*. Serial dependence computed as the peak amplitude of a derivative of Gaussian (DoG) tuning function fitted to the data using least squares is stronger for larger set sizes (see Methods). On the x-axis, 'color of previous trial' refers to the color of the single stimulus probed on the previous trial. (**B–D**) Simulation results for the full model, model with fixed gain parameter $\beta$, and model without plasticity term $w$. Shaded area corresponds to standard error of the mean.

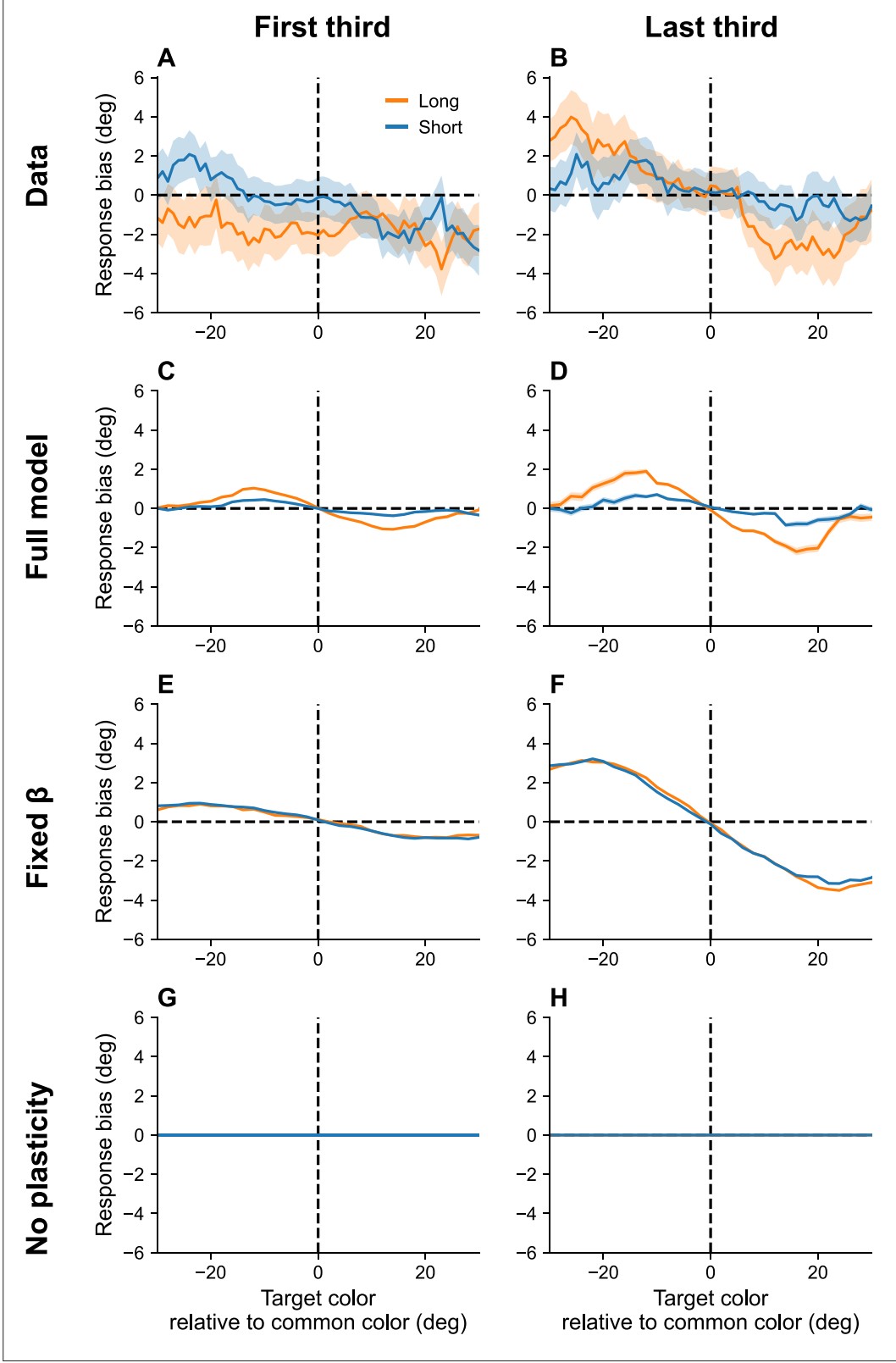

**Figure 7.** Continuous reports are biased toward high-frequency colors. (**A, B**) Bias for targets around common colors during the first (Panel A) and last (Panel B) third of the session, as reported in *Panichello et al., 2019*. Bias refers to the difference between the stimulus and the mean reported color. *x*-Axis is centered around high-frequency colors. Bias increases with RI length (blue = short RI, orange = long RI). Bias also increases as the

*Figure 7 continued on next page*

*Figure 7 continued*

experiment progresses. Simulation results for the full model (**C, D**), model with fixed gain parameter $\beta$ (**E, F**), and model without plasticity term $w$ (**G, H**). Shaded area corresponds to standard error of the mean.

was negatively correlated with the average $\beta$ estimate ($r = -0.32$, p < 0.02; *Figure 8C*). This result is consistent with the hypothesis that dynamic changes in memory performance are associated with changes in neural gain.

## Discussion

We have shown that a simple population coding model with spiking neurons can solve the channel design problem: signals passed through the spiking network are transmitted with close to the minimum achievable distortion under the network's capacity limit. We focused on applying this general model to the domain of working memory, unifying several seemingly disparate aspects of working memory performance: set size effects, stimulus prioritization, serial dependence, approximate timescale invariance, and systematic bias. Our approach builds a bridge between biologically plausible population coding and prior applications of rate-distortion theory to human memory (*Sims et al., 2012*; *Sims, 2015*; *Sims, 2016*; *Bates et al., 2019*; *Bates and Jacobs, 2020*; *Nagy et al., 2020*).

### Relationship to other models

The hypothesis that neural systems are designed to optimize a rate-distortion trade-off has been previously studied through the lens of the information bottleneck method (*Bialek et al., 2006*; *Klampfl et al., 2009*; *Buesing and Maass, 2010*; *Palmer et al., 2015*), a special case of rate-distortion theory in which the distortion function is derived from a compression principle. Specifically, the distortion function is defined as the Kullback–Leibler divergence between $P(\theta'|\theta)$ and $P(\theta'|\hat{\theta})$, where $\theta'$ denotes the probed stimulus. This distortion function applies a 'soft' penalty to errors based on how much probability mass the channel places on each stimulus. The expected distortion is equal to the mutual information between $\theta'$ and $\hat{\theta}$. Thus, the information bottleneck method seeks a channel that maps the input $\theta$ into a compressed representation $\hat{\theta}$ satisfying the capacity limit, while preserving information necessary to predict the probe $\theta'$.

As pointed out by *Leibfried and Braun, 2015*, using the Kullback–Leibler divergence as the distortion function leads to a harder optimization compared to classical rate-distortion theory because $P(\theta'|\hat{\theta})$ depends on the channel distribution, which is the thing being optimized. One consequence of this dependency is that minimizing the rate-distortion objective using alternating optimization (in the style of the Blahut–Arimoto algorithm) is not guaranteed to find the globally optimal channel. It is possible to break the dependency by replacing $P(\theta'|\hat{\theta})$ with a reference distribution that does not depend on the channel. This turns out to strictly generalize rate-distortion theory, because an arbitrary

**Table 1.** Bayesian model comparison between the population coding (PC) model (*Bays, 2014*), the full rate-distortion (RD), and two variants of the RD model (fixed gain and no plasticity). Each model is assigned a protected exceedance probability.

| Experiment | Figure | PC model | RD model (full) | RD model (fixed gain) | RD model (no plasticity) |
|---|---|---|---|---|---|
| *Bays, 2014*, Experiment 1 | 2 | 0.2141 | 0.2286 | 0.4128 | 0.1445 |
| *Bays, 2014*, Experiment 2 | 2 | 0.1853 | 0.7175 | 0.0487 | 0.0485 |
| *Souza and Oberauer, 2015* | 3 | 0.0115 | 0.9785 | 0.0093 | 0.0007 |
| *Bliss and D'Esposito, 2017*, Experiment 1 | 4 | 0.0000 | 1.0000 | 0.0000 | 0.0000 |
| *Bliss et al., 2017*, Experiment 2 | 4 | 0.0029 | 0.7689 | 0.2264 | 0.0018 |
| *Foster et al., 2017* | 5 | 0.3185 | 0.6638 | 0.0089 | 0.0088 |
| *Panichello et al., 2019*, Experiment 1a | 6 | 0.2613 | 0.7387 | 0.0000 | 0.0000 |
| *Panichello et al., 2019*, Experiment 2 | 7 | 0.0544 | 0.9456 | 0.0000 | 0.0000 |

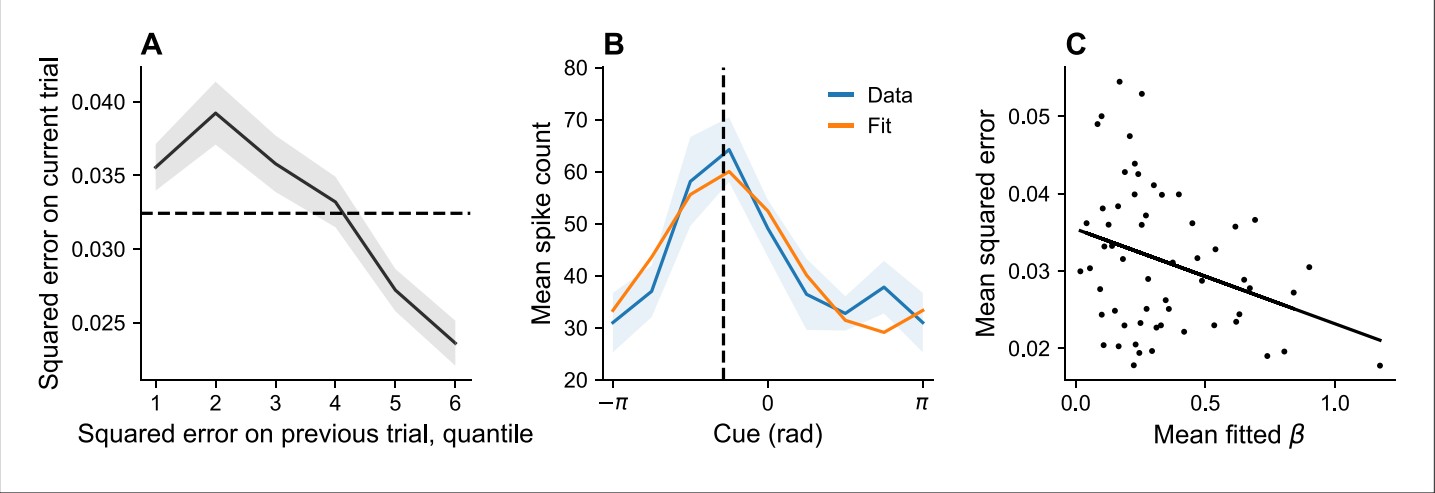

**Figure 8.** Dynamic variation in memory precision and neural gain. (**A**) Mean squared error on current trial, classified by quantiles of squared error on previous trial. Squared error tends to be above average (dashed black line) following low squared error on the previous trial, and tends to be below average following large squared error on the previous trial. (**B**) Angular location tuning curve (orange) fitted to mean spike count (blue) during the retention interval, shown for one example neuron. The neuron's preferred stimulus (dashed black line) corresponds to the peak of the tuning curve. Shaded region corresponds to standard error of the mean. (**C**) Mean squared error for different sessions plotted against mean fitted $\beta$. According to our theory, $\beta$ plays the role of a gain control on the stimulus. Consistent with this hypothesis, memory error decreases with $\beta$.

choice of the reference distribution allows one to recover any lower-bounded distortion function up to a constant offset (*Leibfried and Braun, 2015*). However, existing spiking neuron implementations of the information bottleneck method (*Klampfl et al., 2009*; *Busing and Maass, 2010*) do not make use of such a reference distribution, and hence do not attain the same level of generality.

*Leibfried and Braun, 2015* propose a spiking neuron model that explicitly optimizes the rate-distortion objective function for arbitrary distortion functions. Their approach differs from ours in several ways. First, they model a single neuron, rather than a population. Second, they posit that the channel optimization is realized through synaptic plasticity, in contrast to the intrinsic plasticity rule that we study here. Third, they treat the gain parameter $\beta$ as fixed, whereas we propose an algorithm for optimizing $\beta$.

## Open questions

A cornerstone of our approach is the assumption that the neural circuit responsible for working memory dynamically modifies its output to stay within a capacity limit. What, at a biological level, is the nature of this capacity limit? Spiking activity accounts for a large fraction of cortical energy expenditure (*Attwell and Laughlin, 2001*; *Lennie, 2003*). Thus, a limit on the overall firing rate of a neural population is a natural transmission bottleneck. Previous work on energy-efficient coding has similarly used the cost of spiking as a constraint (*Levy and Baxter, 1996*; *Stemmler and Koch, 1999*; *Balasubramanian et al., 2001*). One subtlety is that the capacity limit in our framework is an upper bound on the stimulus-driven firing rate *relative* to the average firing rate (on a log scale). This means that the average firing rate can be high provided the stimulus-evoked transients are small, consistent with the observation that firing rate tends to be maintained around a set point rather than minimized (*Desai et al., 1999*; *Hengen et al., 2013*; *Hengen et al., 2016*). The set point should correspond to the capacity limit.

The next question is how a neural circuit can control its sensitivity to inputs in such a way that the information rate is maintained around the capacity limit. At the single neuron level, this might be realized by adaptation of voltage conductances (*Stemmler and Koch, 1999*). At the population level, neuromodulators could act as a global gain control. Catecholamines (e.g., dopamine and norepinephrine), in particular, have been thought to play this role (*Servan-Schreiber et al., 1990*; *Durstewitz et al., 1999*). Directly relevant to this hypothesis are experiments showing that local injection of dopamine D1 receptor antagonists into the prefrontal cortex impaired performance in an oculomotor

delayed response task (*Sawaguchi and Goldman-Rakic, 1991*), whereas D1 agonists can improve performance (*Castner et al., 2000*).

In experiments with humans, it has been reported that pharmacological manipulations of dopamine can have non-monotonic effects on cognitive performance, with the direction of the effect depending on baseline dopamine levels (see *Cools and D'Esposito, 2011* for a review). The baseline level (particularly in the striatum) correlates with working memory performance (*Cools et al., 2008*; *Landau et al., 2009*). Taken together, these findings suggest that dopaminergic neuromodulation controls the capacity limit (possibly through a gain control mechanism), and that pushing dopamine levels beyond the system's capacity provokes a compensatory decrease in gain, as predicted by our homeostatic model of gain adaptation. A more direct test of our model would use continuous report tasks to quantify memory precision, bias, and serial dependence under different levels of dopamine.

We have considered a relatively restricted range of visual working memory tasks for which extensive data are available. An important open question concerns the generality of our model beyond these tasks. For example, serial order, AX-CPT, and N-back tasks are widely used but outside the scope of our model. With appropriate modification, the rate-distortion framework can be applied more broadly. For example, one could construct channels for sequences rather than individual items, analogous to how we have handled multiple simultaneously presented stimuli. One could also incorporate a capacity-limited attention mechanism for selecting previously presented information for high fidelity representation, rather than storing everything from a fixed temporal window with relatively low fidelity. This could lead to a new information-theoretic perspective on attentional gating in working memory.

Our model can be extended in several other ways. One, as already mentioned, is to develop a biologically plausible implementation of gain adaptation, either through intrinsic or neuromodulatory mechanisms. A second direction is to consider channels that transmit a compressed representation of the input. Previous work has suggested that working memory representations are efficient codes that encode some stimuli with higher precision than others (*Koyluoglu et al., 2017*; *Taylor and Bays, 2018*). Finally, an important direction is to enable the model to handle more complex memoranda, such as natural images. Recent applications of large-scale neural networks, such as the variational autoencoder, to modeling human memory hold promise (*Nagy et al., 2020*; *Bates and Jacobs, 2020*; *Franklin et al., 2020*; *Bates et al., 2023*; *Xie et al., 2023*), though linking these to more realistic neural circuits remains a challenge.

## Methods

We reanalyzed five datasets with human subjects and one dataset with monkey subjects performing delayed response tasks. The detailed experimental procedures can be found in the original reports (*Bays, 2014*; *Souza and Oberauer, 2015*; *Barbosa et al., 2020*; *Barbosa and Compte, 2020*; *Panichello et al., 2019*; *Bliss and D'Esposito, 2017*). In three of the six datasets, one or multiple colors were presented on a screen at equally spaced locations. After an RI, during which the cues were no longer visible, subjects had to report the color at a particular cued location, measured as angles on a color wheel. In one dataset, angled color bars were presented, and the angle of the bar associated with a cued color had to be reported (*Bays, 2014*). In the two last datasets, only the location of a black cue on a circle had to be remembered and reported (*Barbosa et al., 2020*; *Bliss and D'Esposito, 2017*).

### Set size and stimulus prioritization

Human subjects ($N = 7$) were presented with 2, 4, or 8 color stimuli at the same time. On each trial, one of the locations was cued before the appearance of the stimuli. Cued locations were 3 times as likely to be probed (*Bays, 2014*).

We computed trial-wise error as the circular distance between the reported angle and the target angle, separately for each set size and cuing condition. We then calculated circular variance ($\sigma^2$) and kurtosis ($k$) as presented in the original paper, using the following equations:

$$\sigma^2 = -2 \log |\bar{m}_1|, \tag{18}$$

and

$$k = (|\bar{m}_2| \cos(\text{Arg}(\bar{m}_2) - 2\text{Arg}(\bar{m}_1)) - |\bar{m}_1|^4)/(1 - |\bar{m}_1|)^2, \tag{19}$$

where $\bar{m}_n$ is the $n$ th uncentered trigonometric moment. A histogram with $n = 31$ bins was used to visualize the error distribution in *Figure 2*.

## Timing effects

Human subjects ($N = 36$) were presented with 6 simultaneous color stimuli and had to report the color at a probed location as an angle on a color wheel. The RI and ITI lengths varied across sessions (RI: 1 or 3 s, ITI: 1 or 7.5 s) (*Souza and Oberauer, 2015*). A histogram with $n = 31$ bins was used to visualize the error distribution in *Figure 3*.

## Serial dependence increases with RI and decreases with ITI

Human subjects ($N = 55$) were presented with a black square at a random position on a circle and had to report the location of the cue (*Bliss and D'Esposito, 2017*). The RI and ITI were varied across blocks of trials (RI: 0, 1, 3, 6, or 10 s, ITI: 1, 3, 6, or 10 s). For each block and subject, we computed serial dependence as the peak-to-peak amplitude of a derivative of Gaussian (DoG) function fit to the data. The DoG function is defined as follows:

$$y = xawc \exp(-(wx)^2), \tag{20}$$

where $y$ is the trial-wise error, $x$ is the relative circular distance to the target angle of the previous trial, $a$ is the amplitude of the DoG peak, $w$ is the width of the curve, and $c$ is the constant $\sqrt{2e}$, chosen such that the peak-to-peak amplitude of the DoG fit—the measure of serial dependence in *Bliss and D'Esposito, 2017*—is exactly $2a$.

## Build-up of serial dependence

Human subjects ($N = 12$) performed a delayed continuous report task with one item (*Foster et al., 2017*). Following *Barbosa and Compte, 2020*, we obtained a trial-by-trial measure of serial dependence using their definition of folded error.

Let $\theta_d$ denotes the circular distance between the angle reported on the previous trial and the target angle on the current trial. In order to aggregate trials with negative $\theta_d$ (preceding target is located clockwise to current target) and trials with positive $\theta_d$ (preceding target is located counter-clockwise to current target), we computed the folded error as $\theta_e' = \theta_e \times \text{sign}(\theta_d)$, where $\theta_e$ is the circular distance between the reported angle and the target angle. Positive $\theta_e'$ corresponds to attraction to the previous stimulus, whereas negative $\theta_e'$ corresponds to repulsion.

We excluded trials with absolute errors larger than $\pi/4$. We then computed serial bias as the average folded error in sliding windows of width $\pi/2$ rad and steps of $\pi/30$ rad. We repeated this procedure separately for the trials contained in the first and last third of all sessions. Finally, we computed the increase in serial dependence over the course of a session using a sliding window of 200 trials on the folded error.

## Serial dependence increases with set size

We reanalyzed the dataset collected by *Panichello et al., 2019*, experiment 1a, in which human subjects ($N = 90$) performed a delayed response task with one or three items.

We calculated folded error using the procedure mentioned above. We excluded trials with absolute errors larger than $\pi/4$. We then computed serial bias as the average folded error in sliding windows of width $\pi/4$ rad and steps of $\pi/30$ rad. We repeated this procedure separately for the trials with $M = 1$ or $M = 3$ items. In order to test whether serial dependence was stronger for one of the set size conditions, we performed a permutation test: We shuffled the entire dataset and partitioned it into two groups of size $S_{M=1}$ and $S_{M=3}$, where $S_{M=m}$ denotes the number of trials recorded for the set size condition $M = m$. We fitted a DoG curve (*Equation 20*) to each partition using least squares and computed the difference between the peak amplitude of the two fits. We repeated this process 20,000 times. We then calculated the p-value as the proportion of shuffles for which the difference between the peak amplitudes was equal to or larger than the one computed using the unshuffled dataset.

## Continuous reports are biased toward high-frequency colors

Human subjects ($N = 120$) performed a delayed continuous report task with a set size of 2 (*Panichello et al., 2019*). On each trial, the RI was either 0.5 or 4 s. The stimuli were either drawn from a uniform distribution or from a set of four equally spaced bumps of width $\pi/9$ rad with equal probability. The centers of each bump were held constant for each subject.

We defined systematic bias as mean error versus distance to the closest bump center and computed it in sliding windows of width $\pi/45$ rad and steps of $\pi/90$ rad, as done in the original study. We repeated this procedure separately for the trials with $RI = 0.5s$ or $RI = 4s$, and for the first and last third of trials within a session.

## Simulations and model fitting

For each dataset described above, we performed simulations with three different models: the *full model*, a model with *fixed* $\beta$ ($\alpha = 0$), and a model with *no plasticity* ($\eta = 0$). The following parameters were held fixed for all simulations, unless stated otherwise: $N = 100$, $M = 1$, $\omega = 1$, $\eta = 10^{-3}$, $\alpha = 10^{-1}$, $\Delta t = 5 \times 10^{-2}$ s. Weights $w$ were clipped to be in the range $[-12, 0]$. $\beta$ was initialized at $\beta_0 = 15$ and clipped to be in the range $[0, 1000]$.

In order to account for the higher probing probability of the cued stimulus in *Bays, 2014*, we used

$$\pi_m = \frac{\alpha_m}{\sum_{m'} \alpha_{m'}}, \tag{21}$$

with $\alpha_{\text{priority}} = 3$ and $\alpha_m = 1$ otherwise, as given by the base rates.

Simulations were run on the same trials as given in the dataset. When multiple stimuli were presented simultaneously ($M > 1$) and the values of non-probed stimuli were not included in the dataset, we used stimuli sampled at random in the range $[-\pi, \pi]$ to replace the missing values.

When running a simulation, time was discretized into steps of length $\Delta t$. The simulation time step $\Delta t$ was manually set to provide a good trade-off between simulation resolution and run time. The learning rates $\eta$ and $\alpha$ were scaled by $\Delta t$ to make the simulation results largely independent of the precise choice of $\Delta t$. At each step, spikes $z_i$ were generated by sampling from a Poisson distribution with parameter $\lambda_i = \bar{r} \, r_i \, \Delta t$. Subsequently, $w_i$, $u_i$, $r_i$, $R$, and $\beta$ were computed using the equations given in the main text. At the end of the RI, model predictions were performed by decoding samples generated during a window of $T_d = 0.1$ s using maximum likelihood estimation.

The capacity $C$, the population gain $\bar{r}$, and the plasticity gain parameter $c$ were independently fitted for each subject to maximize the likelihood of the observed errors. To demonstrate the generalizability of these parameter estimates, the parameters were fitted for the dataset from *Bays, 2014* only, and then applied without modification to the other datasets. We used the subject-averaged $C$, $\bar{r}$, and $c$ to run simulations on the remaining datasets. The one exception was for *Souza and Oberauer, 2015*, where responses appeared to be unusually noisy responses; for this dataset, we fixed $C = 0.1$.

In order to compare model performance quantitatively, we fitted the model presented in *Bays, 2014* on the dataset presented in the same paper. This model depends on two free parameters: $\omega$, which controls the tuning width of the neurons, and $\gamma$, which controls the population gain and corresponds to $\bar{r}$ in our text. These parameters were fit to maximize the likelihood of the observed errors; the detailed model fitting procedure can be found in the original report. As outlined above, averaged parameter estimates were used to run simulations on the remaining datasets. Models were subsequently compared by computing the BIC, defined as:

$$\text{BIC} = k \log(n) - 2 \log(L^*), \tag{22}$$

where $k$ is the number of parameters estimated by the model, $n$ is the number of data points, and $L^*$ is the likelihood of the model. For the fitted data, the BIC was used to approximate the marginal likelihood, $P(\text{data}) \approx -\frac{1}{2} \text{BIC}$, which was then submitted to the Bayesian model selection algorithm described in *Rigoux et al., 2014*. Since the same parameters were applied to all the other datasets (i.e., these were generalization tests of the model fit), we instead submitted the log-likelihood directly to Bayesian model selection.

## Dynamics of memory precision and neural gain

We reanalyzed the behavioral and neural dataset collected in *Barbosa et al., 2020*. In this dataset, four adult male rhesus monkeys (*Macaca mulatta*) were trained in an oculomotor delayed response

task that involved fixing their gaze on a central point and subsequently making a saccadic eye movement to the stimulus location after a delay period. While performing the task, firing of neurons in the dorsolateral prefrontal cortex was recorded. Since recordings were not available for all trials within a session, we ignored sessions in which only a subset of the eight potential cues were displayed.

We sorted the squared error on trial $t$ (denoted by $e_t^2$) based on six quantiles of the squared error on the previous trial. We then defined the indicator variable $i_t = \mathcal{I}(e_{t-1}^2 > \bar{e^2})$, taking the value +1 if the squared error on the previous trial was larger than the mean squared error, and −1 otherwise. We then fit the linear mixed model $e_t^2 \sim 1 + i_t + (1|session)$.

In order to infer the preferred stimulus of each recorded neuron, we used a least squares approach to fit the mean spike count for each presented stimulus and neuron to a bell-shaped tuning function:

$$f_i(\theta) = A_i \exp(w_i^{-1}(\cos(\theta - \phi_i) - 1)), \tag{23}$$

where $\theta$ is the presented stimulus, $A_i$ and $w_i$ control the amplitude and width of the tuning function, respectively, and $\phi_i$ is the preferred stimulus of neuron $i$ (**Bays, 2014**).

We then fitted the neural data by performing Poisson regression for each neuron using the following model:

$$\log(s_j) \sim 1 + D_j + \bar{s}_j, \tag{24}$$

where $s_j$ is the number of spikes emitted by the neuron on trial $j$, $D_j$ is the expected distortion between the stimulus $\theta_j$ and the neuron's preferred stimulus, and $\bar{s}_j$ is an exponential moving average of the neuron's spike history with decay rate 0.8. We discarded three neurons for which the fitted $\beta$ was negative and one neuron for which the fitted $\beta$ was larger than 5 standard deviations above the mean of the fitted values.

In order to ascertain the utility of the different regressors, we fitted another model without the history term, and another without both the distortion and history terms, and compared them based on their BIC values.

## Source code

All simulations and analyses were performed using Julia, version 1.6.2. Source code can be found at https://github.com/amvjakob/wm-rate-distortion, (copy archived at **Jakob, 2023**).

## Acknowledgements

Johannes Bill, Wulfram Gerstner, and Chris Bates generously provided constructive feedback and discussion. This research was supported by a Bertarelli Fellowship and by the Center for Brains, Minds, and Machines (funded by NSF STC award CCF-1231216).

## Additional information

### Funding

| Funder | Grant reference number | Author |
| --- | --- | --- |
| Fondation Bertarelli | Bertarelli Fellowship | Anthony MV Jakob |
| National Science Foundation | NSF STC award CCF-1231216 | Samuel J Gershman |

The funders had no role in study design, data collection, and interpretation, or the decision to submit the work for publication.

### Author contributions

Anthony MV Jakob, Conceptualization, Resources, Data curation, Software, Formal analysis, Funding acquisition, Investigation, Visualization, Methodology, Writing – original draft, Writing – review and editing; Samuel J Gershman, Conceptualization, Resources, Supervision, Funding acquisition, Validation, Methodology, Writing – original draft, Project administration, Writing – review and editing

## Author ORCIDs
Anthony MV Jakob  http://orcid.org/0000-0002-0996-1356
Samuel J Gershman  http://orcid.org/0000-0002-6546-3298

### Decision letter and Author response
Decision letter https://doi.org/10.7554/eLife.79450.sa1
Author response https://doi.org/10.7554/eLife.79450.sa2

---

## Additional files

### Supplementary files
• MDAR checklist

### Data availability
The current manuscript is a computational study, so no data have been generated for this manuscript. Source code can be found at https://github.com/amvjakob/wm-rate-distortion (copy archived at *Jakob, 2023*). The previously published datasets are available upon request from the corresponding authors of the published papers, *Souza and Oberauer, 2015*, *Bliss et al., 2017*, and *Panichello et al., 2019*. A minimally processed dataset from *Barbosa et al., 2020* is available online (https://github.com/comptelab/interplayPFC), with the raw data available upon request from the corresponding author of the published paper (raw monkey data available upon request to Christos Constantinidis cconstan@wakehealth.edu, and raw EEG data available upon request to Heike Stein, heike.c.stein@gmail.com). There are no specific application or approval processes involved in requesting these datasets.

The following previously published datasets were used:

| Author(s) | Year | Dataset title | Dataset URL | Database and Identifier |
|---|---|---|---|---|
| Bays P | 2014 | Noise in Neural Populations Accounts for Errors in Working Memory | https://osf.io/s7dhn/ | Open Science Framework, s7dhn |
| Foster J | 2017 | Alpha-band activity reveals spontaneous representations of spatial position in visual working memory | https://osf.io/vw4uc/ | Open Science Framework, vw4uc |
| Barbosa J | 2020 | Interplay between persistent activity and activity-silent dynamics in the prefrontal cortex underlies serial biases in working memory | https://github.com/comptelab/interplayPFC | GitHub, comptelab/interplayPFC |

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
