## [Editor Report]

This important study describes a model neural circuit that learns to optimally represent its inputs subject to an information capacity limit. This novel hypothesis provides a bridge between the theoretical frameworks of rate-distortion theory and neural population coding. Convincing evidence is presented that this model can account for a range of empirical phenomena in the visual working memory literature.

---

## [Decision Letter]

**Decision letter after peer review:**

Thank you for submitting your article "Rate-distortion theory of neural coding and its implications for working memory" for consideration by *eLife*. Your article has been reviewed by 3 peer reviewers, including Paul Bays as the Reviewing Editor and Reviewer #1, and the evaluation has been overseen by Joshua Gold as the Senior Editor.

Essential revisions:

1) Expand and clarify the description of the model and how and why it makes the highlighted predictions, to fill in missing steps and make it more comprehensible to readers from a range of backgrounds.

2) Explain to what extent the neural model and its predictions are compatible with neurophysiological observations of stable tuning functions, and the recurrent excitation believed to maintain activity during a delay.

3) Expand the manuscript to better situate the current model in relation to other mechanistic WM models in the literature, and perform quantitative fitting to more concretely evaluate the model's ability to reproduce empirical data. While formal model comparison (e.g. with AIC) may not be necessary, it is important to evaluate the present model's match to data against other models' to understand its relative strengths and weaknesses. As the model is designed to implement an information capacity limit, its ability to reproduce set size effects (on error variability and error distribution) seems worthy of particular attention.

4) Consider and develop the implications of the model's basic principles (e.g. the idea that periods when WM requirement is low can be traded for a boost in information capacity at a later time) for a broader range of WM tasks and observations. This doesn't have to be quantitative fitting but needs to counter the argument that the empirical data currently presented has been selected to fit the model.

*Reviewer #1 (Recommendations for the authors):*

Some more technical points:

The "simple update rule" of Equation 14 requires knowledge of the information rate R (Equation 13) – is there a plausible method by which this could be computed in the circuit?

WTA, as utilized in the model, is generally a suboptimal decoding method – would predictions change for population vector or ML decoding?

Cosine tuning curves for input neurons lead to von Mises tuning curves for the output neurons – if that's correct it might be worth spelling out as it is the assumed tuning in previous population models of WM.

Excess kurtosis in error distributions has been an important point of debate in WM modelling – it appears to be present in the simulated data, but how does it arise in the model?

How are the smooth "Data" distribution plots generated? Smoothing may be misleading if the shape of the distribution is a question of interest, as it is for set size effects. The smoothing does not appear to take into account the circularity of the response space, based on what happens at π and -pi.

The description of how model predictions are generated needs to be substantially expanded, to explain step-by-step how the data that appears in the "Simulation" panels were produced. How were estimates obtained? How many repetitions/simulated trials were used (the plots are quite noisy)? To what extent were the simulated trials matched to the real ones, in terms of individual stimuli, etc?

What does it mean for the model that the parameter C had to be made ten times smaller for one experiment than the others?

"Spikes contributed to intrinsic synaptic plasticity for 10 timesteps" – what is the implication of this? Is it plausible? Do the results depend on it?

The section on primate data includes results of a model comparison – the methods need to be explained.

Figure 6A axis-label "Relative color of previous trial" – what does this refer to when the set size is three?

The scales in Figure 3B and D don't match.

The variable u_i is in several places (legend of Figure 1, before Equation 15, maybe others) described as the membrane potential instead of excitatory input (which I think is the intended meaning).

Legend of Figure 8B refers to an orientation tuning curve, but I don't believe the task involved oriented stimuli: angular location is probably what is intended.

Figure 1 and elsewhere – using K to indicate set size is going to be very confusing for WM researchers as it has long been used to refer to WM capacity. SS or even N would be preferable.

*Reviewer #2 (Recommendations for the authors):*

Equation (2): More explanation is needed when moving from the constrained optimization problem to the unconstrained optimization by forming a Lagrangian, especially for some readers of *eLife*. Add a couple of sentences justifying this step here.

p.3: OK, I'm going to be a bit nitpicky here. "The rate-distortion function is monotonically decreasing" – I'm not sure you can be sure of this. Increasing the capacity could yield no decrease in distortion in some cases (like if you're already at D=0), so technically you should say "monotonically non-increasing" I think. You also say this a couple of paragraphs later to justify \β being decreased, but I think you can only ensure it's non-increasing, I think, but maybe you can convince me otherwise.

Equation (4): Are you just integrating the ODE defined by (2) and (3) to get this result? If so, say that, or give a bit more info on how you arrive at this formula.

"We assume that inhibition is strong enough such that only one neuron in the population will be active within an infinitesimally small time window." – I think this statement is unnecessary. As long as you make a time window small enough, the probability of co-occurring spikes in a window will be zero.

p.6: When you talk about recurrent models, I don't really see any semblance of recurrence in your models. They're just feedforward, right? The weights w_i seem to really just to be inputs. r_i does not feedback into all the other u_j's. Admittedly, this would be a trickier optimization problem to solve using a simple learning rule. Do you have an idea how you would do this if you introduced recurrence into the formulation Equation. (10) and (11)? You could punt this to the Discussion or get into more detail here, but not worrying enough about the mechanism of maintenance of the population spiking means you have to artificially dial down the capacity as the delay time is increased when you get to the "Timing" section on p.9.

Figure 2: You're qualitatively replicating the data, but admittedly missing some details. Variance scales up slower in the model and kurtosis drops off slower. Why? Is this just a case of people satisficing? Also, I think if you follow the analysis in the supplement of Sims et al. (2012), you should be able to get a good analytic approximation of 2D assuming an unbounded (rather than periodic) domain. Even if you place the problem on a periodic domain, I think you can solve the rate-distortion problem by hand to find the correctly parameterized von Mises distribution.

"Increasing the intertrial interval reduces the information rate, since no stimuli are being communicated during that time period, and can therefore compensate for longer retention intervals." – I don't agree with this interpretation. I would think that longer ITIs might lead to less serial bias. Mechanistically, I don't see what else would disrupt persistent activity in a neural circuit as a function of ITI.

p.10: Neural firing rate activity from the previous trial seems an unlikely candidate for the primary mechanism for serial dependence given the persistent activity is typically gone between trials (see Barbosa et al. 2020) – e.g., short-term plasticity or something else seems more likely. I know you're aiming at a parsimonious model, but I think this point warrants discussion.

For all your Simulation/Data comparisons, it's not clear exactly how you chose model parameters. Did you do something systematic to try and best replicate the trends in data? Did you pick an arbitrary \β and stick with this throughout? More and clearer information about this would be helpful.

*Reviewer #3 (Recommendations for the authors):*

It would be informative to formally compare the current model against alternatives [as one example, Matthey, L., Bays, P. M., and Dayan, P. (2015). A probabilistic palimpsest model of visual short-term memory. PLoS computational biology, 11(1), e1004003.]. As it stands, I am not sure if the model is "yet another model" in a fairly large space, or whether it makes unique predictions or outperforms (or perhaps underperforms) existing models.

Absent formal model comparisons, an extended discussion of the current model's strengths, and especially weaknesses, would greatly improve the manuscript.

[Editors' note: further revisions were suggested prior to acceptance, as described below.]

Thank you for resubmitting your work entitled "Rate-distortion theory of neural coding and its implications for working memory" for further consideration by *eLife*. Your revised article has been evaluated by Joshua Gold (Senior Editor) and a Reviewing Editor.

The manuscript has been improved, and two out of three reviewers are now satisfied, but there are a small number of remaining issues that need to be addressed, as outlined below:

*Reviewer #1 (Recommendations for the authors):*

This revision resolves a number of previous concerns, and the clarity of presentation of the model is much improved. There are two issues remaining that I hope can be dealt with relatively straightforwardly:

1. The revision makes clearer where the empirical data, to which the models are compared and fit, comes from, including that the data in Figure 2A corresponds to Exp 1 in ref [15]. However, if that is the case, why is the data from set size 1 missing? Was it also omitted from model fitting? Please remedy this, as predicting recall performance with a single item is pretty crucial for a working memory model.

2. The problems I noted with data smoothing haven't been dealt with. I checked the plots in the original papers corresponding to Figures2 and 3, and its clear the smoothing is strongly distorting the distribution shapes (which form a key part of the evidence the models are intended to reproduce). Is there a reason you can't just plot histograms, like the original papers did for the data?

---

## [Author Response]

Essential revisions:1) Expand and clarify the description of the model and how and why it makes the highlighted predictions, to fill in missing steps and make it more comprehensible to readers from a range of backgrounds.

We’ve now expanded the methods and Results sections to make the model description clearer. Please see responses to the comments below.

2) Explain to what extent the neural model and its predictions are compatible with neurophysiological observations of stable tuning functions, and the recurrent excitation believed to maintain activity during a delay.

As we explain below and on p. 7, we deliberately chose to be agnostic about the memory maintenance mechanism. Several have been proposed in the literature, all of which might be compatible with our framework. The essential constraint imposed by our framework is the capacity limit. Any maintenance mechanism that adheres to this capacity limit is compatible.

We’re not entirely sure what aspect of stable tuning is being referenced here. Some studies suggest representational drift in visual working memory (e.g., Murray et al., 2017; Wolff et al., 2020), so it’s a bit unclear how stable the tuning functions really are. In our setup, the input and output neurons have stable tuning functions; what changes across time is only the gain and excitability.

3) Expand the manuscript to better situate the current model in relation to other mechanistic WM models in the literature, and perform quantitative fitting to more concretely evaluate the model's ability to reproduce empirical data. While formal model comparison (e.g. with AIC) may not be necessary, it is important to evaluate the present model's match to data against other models' to understand its relative strengths and weaknesses. As the model is designed to implement an information capacity limit, its ability to reproduce set size effects (on error variability and error distribution) seems worthy of particular attention.

We now include quantitative model fitting and comparison (summarized in Table 1), both to an existing model (Bays, 2014) and to several variants of the rate-distortion model that lack key features (gain adaptation and intrinsic plasticity).

4) Consider and develop the implications of the model's basic principles (e.g. the idea that periods when WM requirement is low can be traded for a boost in information capacity at a later time) for a broader range of WM tasks and observations. This doesn't have to be quantitative fitting but needs to counter the argument that the empirical data currently presented has been selected to fit the model.

We believe that the standard for breadth should be based on comparable papers in the literature. The most closely related papers to our own are Bays (2014) and Sims (2015). Both of those papers analyzed similar visual working memory tasks (indeed, we analyze some of the same datasets), using similar visual and quantitative metrics. Neither paper attempted to model neurophysiological data, serial dependence, stimulus inhomogeneities, or the effect of intertrial interval timing (some of these were addressed in later papers, though). Given the breadth of findings that are already modeled in our paper, we think that it’s not really a fair assessment to say that we selected data to fit the model. We are in fact trying to explain data that were ignored by many previous models.

We agree that expanding the broader implications of our research is a useful addition. We have done this in a few places. In the “open questions” section of the Discussion (p. 19), we have added the following paragraph:

“In experiments with humans, it has been reported that pharmacological manipulations of dopamine can have non-monotonic effects on cognitive performance, with the direction of the effect depending on baseline dopamine levels (see [83] for a review). The baseline level (particularly in the striatum) correlates with working memory performance [84, 85]. Taken together, these findings suggest that dopaminergic neuromodulation controls the capacity limit (possibly through a gain control mechanism), and that pushing dopamine levels beyond the system’s capacity provokes a compensatory decrease in gain, as predicted by our homeostatic model of gain adaptation. A more direct test of our model would use continuous report tasks to quantify memory precision, bias, and serial dependence under different levels of dopamine.”

Immediately below this paragraph, we have added another new paragraph addressing potential extensions that can address a wider range of tasks:

“We have considered a relatively restricted range of visual working memory tasks for which extensive data are available. An important open question concerns the generality of our model beyond these tasks. For example, serial order, AX-CPT, and N-back tasks are widely used but outside the scope of our model. With appropriate modification, the rate-distortion framework can be applied more broadly. For example, one could construct channels for sequences rather than individual items, analogous to how we have handled multiple simultaneously presented stimuli. One could also incorporate a capacity-limited attention mechanism for selecting previously presented information for high fidelity representation, rather than storing everything from a fixed temporal window with relatively low fidelity. This could lead to a new information-theoretic perspective on attentional gating in working memory.”

Reviewer #1 (Recommendations for the authors):Some more technical points:The "simple update rule" of Equation 14 requires knowledge of the information rate R (Equation 13) – is there a plausible method by which this could be computed in the circuit?

This is a great question. We have sketched an answer on p. 6:

“In this paper, we do not directly model how the information rate R is estimated in a biologically plausible way. One possibility is that this is implemented with slowly changing extracellular calcium levels, which decrease when cells are stimulated and then slowly recover. This mirrors (inversely) the qualitative behavior of the information rate. More quantitatively, it has been posited that the relationship between firing rate and extracellular calcium level is logarithmic [40], consistent with the mathematical definition in Equation. 13. Thus, in this model, capacity C corresponds to a calcium set point, and the gain parameter adapts to maintain this set point. A related mechanism has been proposed to control intrinsic excitability via calcium-driven changes in ion channel conductance [41, 42].”

WTA, as utilized in the model, is generally a suboptimal decoding method – would predictions change for population vector or ML decoding?

This is an interesting question, but note that in this case the WTA mechanism is in fact optimal under our specified objective function. We think comparing different decoders would take us too far afield, since our goal was to show how to implement an optimal channel under rate-distortion theory.

Cosine tuning curves for input neurons lead to von Mises tuning curves for the output neurons – if that's correct it might be worth spelling out as it is the assumed tuning in previous population models of WM.

Thanks for pointing this out. We now note this explicitly on p. 7.

Excess kurtosis in error distributions has been an important point of debate in WM modelling – it appears to be present in the simulated data, but how does it arise in the model?

Excess kurtosis was noted by Bays (2014) in his population coding model, but as far as we know there isn’t a naturally intuitive explanation for why this arises from the model. Although we don’t have an intuitive explanation, we’ve elaborated our description of the phenomenon on p. 9:

“Kurtosis is one way of quantifying deviation from normality: values greater than 0 indicate tails of the error distribution that are heavier than expected under a normal distribution. The ‘excess’ kurtosis observed in our model is comparable to that observed by Bays in his population coding model [15] when gain is sufficiently low. This is not surprising, given the similarity of the models.”

How are the smooth "Data" distribution plots generated? Smoothing may be misleading if the shape of the distribution is a question of interest, as it is for set size effects. The smoothing does not appear to take into account the circularity of the response space, based on what happens at π and -pi.

For Figures 2 and 3, we used a kernel density estimate plot to visualize the smoothed data distribution. We used the Python function `kdeplot` from the `seaborn` package with the default parameters.

For Figures 5, 6, 7, we obtained the smoothed data distribution plots by moving window averaging. The precise smoothing parameters we used are given in the Methods. While smoothing the data distribution might introduce artifacts on the edges of the plot domain, the rest of the plot is unaffected for sufficiently light smoothing.

The description of how model predictions are generated needs to be substantially expanded, to explain step-by-step how the data that appears in the "Simulation" panels were produced. How were estimates obtained? How many repetitions/simulated trials were used (the plots are quite noisy)? To what extent were the simulated trials matched to the real ones, in terms of individual stimuli, etc?

We have clarified the model prediction procedure in a step-by-step fashion on p. 22. Furthermore, in the revised version of this manuscript, the simulated trials exactly correspond to the trials in the dataset. While the investigated datasets always contained information on the probed stimulus, in the case of multiple simultaneous stimuli the value of non-probed stimuli were sometimes omitted. In such cases, the missing values were replaced with stimuli sampled at random in the range [-pi, pi].

What does it mean for the model that the parameter C had to be made ten times smaller for one experiment than the others?

As we now clarify in the paper, this particular dataset appears to contain unusually noisy responses, which might be due to several factors, such as the task being more challenging or the subjects being less attentive. Therefore, in order to reasonably fit the dataset, we lowered the model gain. This modeling choice is obviously ad hoc, but hopefully it is justified based on the adequacy of the model fit.

"Spikes contributed to intrinsic synaptic plasticity for 10 timesteps" – what is the implication of this? Is it plausible? Do the results depend on it?

Our results don’t depend strongly on this assumption, and it has been removed from the manuscript. In the revised model, a spike contributes to intrinsic plasticity for one timestep only, the timestep during which it was generated.

The section on primate data includes results of a model comparison – the methods need to be explained.

We have expanded the methods section on primate data analysis, including information on the source of the neural recordings in addition to details of model comparison.

Figure 6A axis-label "Relative color of previous trial" – what does this refer to when the set size is three?

Despite showing three cues, only one is probed during each trial. In this context, “relative color of previous trial” refers to the color of the cue that was probed on the previous trial. We now state this explicitly in the caption.

The scales in Figure 3B and D don't match.

Thank you for pointing this out; it has been corrected.

The variable u_i is in several places (legend of Figure 1, before Equation 15, maybe others) described as the membrane potential instead of excitatory input (which I think is the intended meaning).

Thanks for catching that error; we’ve corrected it throughout.

Legend of Figure 8B refers to an orientation tuning curve, but I don't believe the task involved oriented stimuli: angular location is probably what is intended.

Corrected.

Figure 1 and elsewhere – using K to indicate set size is going to be very confusing for WM researchers as it has long been used to refer to WM capacity. SS or even N would be preferable.

We’ve changed “K” to “M” throughout.

Reviewer #2 (Recommendations for the authors):Equation (2): More explanation is needed when moving from the constrained optimization problem to the unconstrained optimization by forming a Lagrangian, especially for some readers of eLife. Add a couple of sentences justifying this step here.

We’ve added several sentences around Equation. 2 to hopefully make this more accessible.

p.3: OK, I'm going to be a bit nitpicky here. "The rate-distortion function is monotonically decreasing" – I'm not sure you can be sure of this. Increasing the capacity could yield no decrease in distortion in some cases (like if you're already at D=0), so technically you should say "monotonically non-increasing" I think. You also say this a couple of paragraphs later to justify \β being decreased, but I think you can only ensure it's non-increasing, I think, but maybe you can convince me otherwise.

We’ve changed “decreasing” to “non-increasing” in both places.

Equation (4): Are you just integrating the ODE defined by (2) and (3) to get this result? If so, say that, or give a bit more info on how you arrive at this formula.

We are indeed integrating the ODE defined by Eqs. 2 and 3 to obtain Equation. 4. We have clarified this on p. 22.

"We assume that inhibition is strong enough such that only one neuron in the population will be active within an infinitesimally small time window." – I think this statement is unnecessary. As long as you make a time window small enough, the probability of co-occurring spikes in a window will be zero.

We have removed this sentence.

p.6: When you talk about recurrent models, I don't really see any semblance of recurrence in your models. They're just feedforward, right? The weights w_i seem to really just to be inputs. r_i does not feedback into all the other u_j's. Admittedly, this would be a trickier optimization problem to solve using a simple learning rule. Do you have an idea how you would do this if you introduced recurrence into the formulation Equation. (10) and (11)? You could punt this to the Discussion or get into more detail here, but not worrying enough about the mechanism of maintenance of the population spiking means you have to artificially dial down the capacity as the delay time is increased when you get to the "Timing" section on p.9.

We only mention recurrent networks as one possible model that realizes persistent activation (on p. 7 we discuss other possibilities). We didn’t want to commit to a particular implementation because our theoretical framework is largely agnostic with respect to this choice (this is arguably true of the Bays 2014 population coding model as well).

Note that we did not artificially dial down the capacity as the delay time increases. The gain parameter is endogenously set by Equation. 14. Critically, capacity is assumed to stay fixed across time (we treat this as a physical/structural property of the memory system). As explained in the “memory maintenance” section, interval dependence arises from the fact that a fixed capacity is allocated (typically evenly) across a given interval.

Figure 2: You're qualitatively replicating the data, but admittedly missing some details. Variance scales up slower in the model and kurtosis drops off slower. Why? Is this just a case of people satisficing? Also, I think if you follow the analysis in the supplement of Sims et al. (2012), you should be able to get a good analytic approximation of 2D assuming an unbounded (rather than periodic) domain. Even if you place the problem on a periodic domain, I think you can solve the rate-distortion problem by hand to find the correctly parameterized von Mises distribution.

With respect to the figure, it’s true that we are missing some details, but in our view the differences between model and data are rather subtle.

The analysis suggestion is intriguing, but we felt that it is outside the scope of our paper, which tries to follow closely the setup of Bays (2014), which assumed a periodic domain.

"Increasing the intertrial interval reduces the information rate, since no stimuli are being communicated during that time period, and can therefore compensate for longer retention intervals." – I don't agree with this interpretation. I would think that longer ITIs might lead to less serial bias. Mechanistically, I don't see what else would disrupt persistent activity in a neural circuit as a function of ITI.

Maybe there is some confusion here? The data show that longer ITIs reduce serial dependence, and we reproduce this with our model. So we’re not sure what the point of disagreement is.

p.10: Neural firing rate activity from the previous trial seems an unlikely candidate for the primary mechanism for serial dependence given the persistent activity is typically gone between trials (see Barbosa et al. 2020) – e.g., short-term plasticity or something else seems more likely. I know you're aiming at a parsimonious model, but I think this point warrants discussion.

To be clear, we are not suggesting that the model is updating the gain parameter on a trial-by-trial basis. Rather, Equation. 14 is being applied continuously (which we now state explicitly). So there is no need to assume that persistent activity is maintained between trials.

For all your Simulation/Data comparisons, it's not clear exactly how you chose model parameters. Did you do something systematic to try and best replicate the trends in data? Did you pick an arbitrary \β and stick with this throughout? More and clearer information about this would be helpful.

Thanks for this suggestion. We have now completely redone the modeling using fitted parameters. Our model fitting procedures are described in the Methods (p. 22).

Reviewer #3 (Recommendations for the authors):It would be informative to formally compare the current model against alternatives [as one example, Matthey, L., Bays, P. M., and Dayan, P. (2015). A probabilistic palimpsest model of visual short-term memory. PLoS computational biology, 11(1), e1004003.]. As it stands, I am not sure if the model is "yet another model" in a fairly large space, or whether it makes unique predictions or outperforms (or perhaps underperforms) existing models.

This is a very interesting model of visual working memory. However, it is primarily focused on memory for multi-feature items, a topic that we don’t address in this paper. While the binding problem in memory is extremely important, we felt that it was outside the scope of this paper, especially given that we are comparing our model to several others already.

Absent formal model comparisons, an extended discussion of the current model's strengths, and especially weaknesses, would greatly improve the manuscript.

Agreed, which is why now we include formal model comparison, summarized in Table 1 and discussed on p. 14.

[Editors' note: further revisions were suggested prior to acceptance, as described below.]

The manuscript has been improved, and two out of three reviewers are now satisfied, but there are a small number of remaining issues that need to be addressed, as outlined below:Reviewer #1 (Recommendations for the authors):This revision resolves a number of previous concerns, and the clarity of presentation of the model is much improved. There are two issues remaining that I hope can be dealt with relatively straightforwardly:1. The revision makes clearer where the empirical data, to which the models are compared and fit, comes from, including that the data in Figure 2A corresponds to Exp 1 in ref [15]. However, if that is the case, why is the data from set size 1 missing? Was it also omitted from model fitting? Please remedy this, as predicting recall performance with a single item is pretty crucial for a working memory model.

Thank you for raising this point. We included the data from set size 1 in the model fitting; we merely omitted plotting it in Figure 2A for visual consistency with Figure 2BC, which uses the dataset from Exp 2 in ref [15], for which only set sizes 2, 4 and 8 are available.

We have now additionally plotted the data and simulation results pertaining to set size 1 in Figure 2A,D,G,J.

2. The problems I noted with data smoothing haven't been dealt with. I checked the plots in the original papers corresponding to Figures2 and 3, and its clear the smoothing is strongly distorting the distribution shapes (which form a key part of the evidence the models are intended to reproduce). Is there a reason you can't just plot histograms, like the original papers did for the data?

We initially smoothed the error distribution plots to remove sampling noise from the simulation results. However, as rightfully noted, this was distorting the distribution shape, especially at the boundaries of the function domain.

We have remedied this matter by plotting histograms instead for both the original data and the simulation results in Figure 2 and Figure 3.